# A global-temporal analysis on *Phytophthora sojae* resistance-gene efficacy

Austin G. McCoy [1] ✉, Richard R. Belanger [2], Carl A. Bradley [3], Daniel G. Cerritos-Garcia [4], Vinicius C. Garnica [5], Loren J. Giesler[6], Pablo E. Grijalba[7], Eduardo Guillin[8], Maria A. Henriquez[9], Yong Min Kim [10], Dean K. Malvick [11], Rashelle L. Matthiesen [12], Santiago X. Mideros [13], Zachary A. Noel [14], Alison E. Robertson[12], Mitchell G. Roth [15], Clarice L. Schmidt[12], Damon L. Smith [16], Adam H. Sparks [17,18], Darcy E. P. Telenko [19], Vanessa Tremblay[2], Owen Wally [20] & Martin I. Chilvers [1] ✉

Plant disease resistance genes are widely used in agriculture to reduce disease outbreaks and epidemics and ensure global food security. In soybean, *Rps* (Resistance to *Phytophthora sojae*) genes are used to manage *Phytophthora sojae*, a major oomycete pathogen that causes Phytophthora stem and root rot (PRR) worldwide. This study aims to identify temporal changes in *P. sojae* pathotype complexity, diversity, and *Rps* gene efficacy. Pathotype data was collected from 5121 isolates of *P. sojae*, derived from 29 surveys conducted between 1990 and 2019 across the United States, Argentina, Canada, and China. This systematic review shows a loss of efficacy of specific *Rps* genes utilized for disease management and a significant increase in the pathotype diversity of isolates over time. This study finds that the most widely deployed *Rps* genes used to manage PRR globally, *Rps1a*, *Rps1c* and *Rps1k*, are no longer effective for PRR management in the United States, Argentina, and Canada. This systematic review emphasizes the need to widely introduce new sources of resistance to *P. sojae*, such as *Rps3a*, *Rps6*, or *Rps11*, into commercial cultivars to effectively manage PRR going forward.

Soybean (*Glycine max* L.) is a major source of protein and oil that is produced on approximately 130 million hectares globally each year[1]. As the global population increases from 7.9 billion to an estimated 9.8 billion by 2050, we will need to produce more food on less land to ensure global food security[2]. The mission to increase the seed yield potential of soybeans has been at the forefront of soybean breeders for decades, and consequently, present soybean varieties produce nearly twice the seed yield they did 60 years ago[3]. However, efforts to achieve yield gains through genetics are often at the expense of decreased levels of disease resistance[4].

[1]Michigan State University, East Lansing, MI, USA. [2]Université Laval, Québec, Canada. [3]University of Kentucky, Princeton, KY, USA. [4]University of Connecticut, Storrs Mansfield, CT, USA. [5]North Carolina State University, Raleigh, NC, USA. [6]University of Nebraska-Lincoln, Lincoln, NE, USA. [7]Universidad de Buenos Aires, Buenos Aires, Argentina. [8]Instituto Nacional de Tecnologia Agropecuaria, Buenos Aires, Argentina. [9]Agriculture and Agri-Food Canada, Morden, Manitoba, Canada. [10]Agriculture and Agri-Food Canada, Brandon, Manitoba, Canada. [11]University of Minnesota, St Paul, MN, USA. [12]Iowa State University, Ames, IA, USA. [13]University of Illinois at Urbana-Champaign, Urbana, IL, USA. [14]Auburn University, Auburn, AL, USA. [15]The Ohio State University-Wooster, Wooster, OH, USA. [16]University of Wisconsin-Madison, Madison, WI, USA. [17]Department of Primary Industries and Regional Development, Perth, WA, Australia. [18]University of Southern Queensland, Toowoomba, Qld, Australia. [19]Purdue University, West Lafayette, IN, USA. [20]Agriculture and Agri-Food Canada, Harrow, ON, Canada. ✉e-mail: mccoyaus@msu.edu; chilvers@msu.edu

Phytophthora stem and root rot (PRR) of soybean, caused by the soilborne oomycete *Phytophthora sojae* (Kaufmann & Gerdemann), is responsible for 1–2 billion U.S. dollars in soybean yield loss worldwide[5]. Outbreaks of PRR are more prevalent after heavy rains in fields where soil moisture remains saturated and facilitates the production of motile zoospores of *P. sojae*, which move chemotactically toward soybean root exudates and infect soybean roots. During infection, the sexual reproductive and resting structure, the oospore, is produced within infected soybean root and stem tissue[6]. *P. sojae* is homothallic, or self-fertile, and can produce viable oospores without the need for a second genotype. Oospores of *P. sojae* can lie dormant within plant debris in the soil for years until environmental conditions are conducive to infection (i.e., flooding) and susceptible host plants are present[6,7].

Current climate models suggest that periods of intense rain may become more frequent due to anthropogenically driven climate change[8]. The increasing frequency of heavy rains could provide more favorable environmental conditions for PRR disease throughout the global soybean-producing regions, emphasizing that effective and economically viable management of this destructive soybean disease will be increasingly important[9].

Currently, soybean varieties with single-dominant *P. sojae* resistance genes (termed "*Rps*" genes, "resistance to *Phytophthora sojae*") and quantitative resistance are commercially available to mitigate losses to PRR specifically. The first *Rps* gene available for PRR management was *Rps1a*, released in 1964, followed by *Rps1c* in 1980, and lastly, *Rps1k* and *Rps3a*, released in 1985[10]. The genes *Rps1c*, *Rps1k*, and *Rps3a* were made commercially available to manage PRR after *Rps1a*-mediated resistance in Ohio became no longer effective in 1972[10]. Given the adaptation of the *P. sojae* population to evade *Rps1a*-mediated resistance over 8 years, Schmitthenner hypothesized that *Rps* genes would only be effective for management for 6 to 15 years before needing to be replaced by more efficacious *Rps* genes[11]. This observation raises current concerns for the efficacy and longevity of *Rps1c*, *Rps1k*, and *Rps3a* for PRR management on a global scale, as they have been deployed for 35–40 years[10]. Currently, more than 40 *Rps* genes have been identified in soybean germplasm; however, only a few are widely effective and released for PRR management[6,12]. In North and South America, *Rps1c* and *Rps1k* are the most widely available *Rps* genes to farmers for PRR management, while the availability of *Rps1a*, *Rps3a*, *Rps6*, and *Rps8* depends on country, locale, and seed company[6,13].

Soybean *Rps* proteins initiate strong qualitative defense responses against *P. sojae* isolates expressing specific *Avr* proteins (effectors). This gene-for-gene relationship has been well characterized and is the primary means of disease management for PRR in agricultural production worldwide[6,13–16]. Soybean *Rps* genes are thought to encode for NBS-LRR (Nucleotide Binding Site–Leucine Rich Repeat) receptor proteins which recognize *Avr* proteins or their activity from *P. sojae*. When *P. sojae* produces *Avr* proteins detected by the complementary soybean *Rps* protein(s), genetic signals are produced that lead to a hyper-sensitive response that confers complete resistance to *P. sojae*[17]. Alternatively, *P. sojae* isolates that do not express effectors recognized by the cognate soybean *Rps* protein evade detection, continue infection, and cause PRR. A standard set of soybean germplasm, which differs in the *Rps* gene present in the genome, is used to characterize the gene-for-gene interaction phenotypically and identify effective *Rps* genes in the sampled *P. sojae* population[18]. Recently, new molecular tools have been developed to rapidly identify *Avr* genes present in *P. sojae* isolates and thus assess the pathotype profile without going through the phenotypic procedures[19,20]. Through surveying populations of *P. sojae* and describing their pathogenicity phenotype, effective resistance genes for the management of the sampled population can be identified and recommended for use by commercial producers.

Regional or state sampling of *P. sojae* populations has occurred in North America since the first pathotype of *P. sojae* was described in 1965[11,14,21–41]. While *P. sojae* was identified and known to be prevalent in South America since the 1970s, surveys have only been conducted since the late 1980s[15,42–46]. Surveys in Asia, primarily China, where *P. sojae* is an invasive pest, have been performed since 1991, when the pathogen was first identified in Heilongjiang Province, China[16,47–54]. Surveys in other soybean-growing countries such as Japan, Iran, and Australia have been infrequently conducted[55–58]. These surveys typically were performed on a state or province scale and occasionally aggregated to offer insights into *Rps* gene efficacy on a regional scale at a single time point[13,14,38,49]. In addition to identifying effective *Rps* genes, these studies have also been used to describe and track the pathotype complexity, the number of *Rps* genes which isolates can cause disease against, as well as the diversity of pathotypes to better understand the durability of *Rps*-mediated resistance. As the pathotype complexity of the *P. sojae* population increases, there will be fewer individual *Rps* genes that can effectively manage the known population and stacking, or having multiple *Rps* genes present within the genotype, will be needed for effective disease management. Likewise, observations of increasingly diverse pathotypes mean it is highly unlikely these populations would be effectively managed through the use of a single *Rps* gene. Pathotype complexity and diversity metrics can be used to assess the potential durability of employed *Rps* genes in a given population. Pathotype evaluation surveys at multiple timeframes in the same regions have increased concerns about increasing pathotype complexity and diversity, and the potential loss of effectiveness of the *Rps1c* and *Rps1k* genes for PRR management in North and South America have emerged[14,36–39,44,45]. However, no comprehensive worldwide analysis has been performed to evaluate the durability of *Rps* genes over time and across continents, which would allow for more informed planning and decision-making in global commercial trait-introduction programs to adequately manage PRR in the field.

Identifying how regional *P. sojae* pathotypes have evolved, as well as the durability of *Rps* genes for PRR management, will help guide soybean breeding and disease management recommendations on a global scale. Here, we use a novel approach of collating data from the past thirty years of *P. sojae* pathotype surveys from four of the largest soybean-producing countries (Argentina, Canada, China, and the United States) across three continents to show the first global-scale *Rps* gene fluctuation. Specifically, we aimed to: (1) identify countrywide-temporal changes in pathotype complexity, (2) evaluate *Rps* gene efficacy on a country-temporal scale, and (3) determine how the pathotype diversity of sampled *P. sojae* populations have changed over time by country. We determined that: (1) the pathotype complexity of *P. sojae* populations in Argentina, China, and the United States has significantly increased over time, with sampled populations now able to overcome 1–3 more of the tested *Rps* genes than in previously sampled time frames, (2) *Rps1c* and *Rps1k* are no longer effective in the United States, Argentina, and Canada; however there is little change in *Rps1c* and *Rps1k* efficacy observed in China between the time frames studied, and (3) the diversity of pathotypes has significantly changed over time in each country, leading to inadequate PRR disease management using currently employed *Rps* genes. Together, these results broadly build upon foundational research on the management of PRR. Moreover, they significantly improve our current understanding of *Rps* gene efficacy over time and provide a rationale for deploying new *Rps* genes and quantitative resistance to ensure adequate management of PRR globally.

## Results

*Phytophthora sojae* pathotype composition within populations is thought to evolve and change over time due in part to the selective

**Table 1 | Studies used in this systematic review**

| Study | State or Province | Country | Years sampled | Isolate recovery method[a] | Pathotype evaluation method[b] | Time frame grouping |
|---|---|---|---|---|---|---|
| Schmitthenner, Hobe, and Bhat[11] | Ohio | United States | 1978–1991[c] | Plant isolations and soil baiting | Hypocotyl inoculation | 1990–1999 |
| Yang et al.[27] | Iowa | United States | 1991–1994 | Plant isolations and soil baiting | Hypocotyl inoculation | 1990–1999 |
| Abney et al.[28] | Indiana | United States | 1993 | Plant isolations and soil baiting | Hypocotyl inoculation | 1990–1999 |
| Kaitany et al.[29] | Michigan | United States | 1993–1997 | Plant isolations | Hypocotyl inoculation | 1990–1999 |
| Dorrance et al.[31] | Ohio | United States | 1997–1999 | Soil baiting | Hypocotyl inoculation | 1990–1999 |
| Jackson et al.[30] | Arkansas | United States | 1995–1998 | Soil baiting | Hypocotyl inoculation | 1990–1999 |
| Malvick and Grunden[32] | Illinois | United States | 2001–2002 | Soil baiting | Hypocotyl inoculation | 2000–2013 |
| Nelson et al.[33] | North Dakota | United States | 2002–2004 | Soil baiting | Hypocotyl inoculation | 2000–2013 |
| Robertson et al.[34] | Iowa | United States | 2005 | Soil baiting | Hypocotyl inoculation | 2000–2013 |
| Dorrance et al.[14] | 11 U.S states | United States | 2012–2013 | Plant isolations and soil baiting | Hypocotyl inoculation | 2000–2013 |
| Hebb et al.[38] | Illinois, Indiana, Kentucky, Ohio | United States | 2016–2018 | Soil baiting | Hypocotyl inoculation | 2013–2019 |
| Chowdhury et al.[36] | South Dakota | United States | 2013–2017 | Soil baiting | Hypocotyl inoculation | 2013–2019 |
| McCoy et al.[39] | Michigan | United States | 2017 | Soil baiting | Hypocotyl inoculation | 2013–2019 |
| Matthiesen et al.[37] | Nebraska, Iowa | United States | 2016–2018 | Plant isolations and soil baiting[d] | Hypocotyl inoculation | 2013–2019 |
| Barreto et al.[42] | Buenos Aires | Argentina | 1989–1992 | Plant isolations and soil baiting | Hypocotyl inoculation | 1989–1999 |
| Grijalba et al.[15] | Pampeana subregion | Argentina | 1998–2004[c] | Plant isolations and soil baiting | Hypocotyl inoculation | 1989–1999/ 2000–2013 |
| Grijalba et al.[44] | Buenos Aires | Argentina | 2013–2015 | Plant isolations and soil baiting | Hypocotyl inoculation | 2014–2019 |
| Grijalba et al.[45] | Pampas region | Argentina | 2013–2015 | Plant isolations and soil baiting | Hypocotyl inoculation | 2014–2019 |
| Xue et al.[40] | Ontario | Canada | 2010–2012 | Soil baiting | Hypocotyl inoculation | 2000–2013 |
| Henriquez et al.[41] | Manitoba | Canada | 2014–2017 | Plant isolations | Hypocotyl inoculation | 2014–2019 |
| Tremblay et al.[13] | Quebec, Manitoba, Ontario | Canada | 2016–2019 | Soil baiting | qPCR assay for Avr genes | 2014–2019 |
| Jingzhi et al.[48] | Heilongjiang, Jilin | China | 1990–1999 | Plant isolations | Hypocotyl inoculation | 1990–1999 |
| Zhu et al.[49] | Heilongjiang, Jilin, Inner Mongolia, Anhui, Henan, Shandong, Jiangsu, Zhejiang | China | 1990–1999[c] | Plant isolations and soil baiting | Hypocotyl inoculation | 1990–1999/ 2000–2013 |
| Xiuhong et al.[50] | Heilongjiang, Jilin | China | 1990–1999 | Plant isolations and soil baiting | Hypocotyl inoculation | 1990–1999 |
| Zhang et al.[51] | Jilin | China | 2007–2015 | Soil baiting | Hypocotyl inoculation | 2000–2013 |
| Cui et al.[52] | Heilongjiang, Fujian | China | 2007 | Soil baiting | Hypocotyl inoculation | 2000–2013 |
| Linkai et al.[53] | Xinjiang | China | 2007–2008 | Soil baiting | Hypocotyl inoculation | 2000–2013 |
| Tian et al.[54] | Heilongjiang | China | 2011–2015 | Soil baiting | Hypocotyl inoculation | 2000–2013 |
| Zhang et al.[16] | Jilin | China | 2007–2015 | Soil baiting | Hypocotyl inoculation | 2000–2013 |

[a]Plant isolations refers to isolations directly from field-collected plants. Soil baiting refers to soil samples having been collected and a susceptible soybean genotype used to "bait" pathogen infection and then isolations performed.

[b]Hypocotyl inoculation refers to the method described in Dorrance et al.[68] in which soybean seedlings containing a single *Rps*-gene are inoculated with an isolate, and the interaction is rated based on disease development or absence of disease development. qPCR assay for Avr genes refers to the method in Tremblay et al.[13] in which *P. sojae* genomic DNA is used to detect the Avr genes that interact with *Rps* genes.

[c]Isolates in this study were used between two time points, depending on when individual isolates were acquired. No single isolate was used at more than one time point. No data prior to 1990 was used in this study.

[d]Iowa used the soil baiting technique; Nebraska used plant isolations and soil baiting techniques.

pressure of always planting soybean varieties containing the same *Rps* gene each rotation[10]. A *P. sojae* population adapts due to this constant selective pressure from the host through mechanisms such as out-crossing with other *P. sojae* genotypes[59], as well as mutations[60,61] or gene-silencing[62] of the detected *P. sojae Avr*-gene product. If the soybean *Rps* gene is unable to detect infection by *P. sojae*, no defense response is activated; the plant is therefore susceptible to infection, and disease ensues. In total, 41 *P. sojae* pathotype studies were identified through our search of the literature. Of these, 29 were used for this systematic review (Table 1; Supplementary Fig. 6). The remaining 12 studies were not included as they were either conducted outside of the time frames studied, did not test the standard soybean *Rps* genes

used in pathotyping, or the country in which the study took place did not have sufficient additional surveys for temporal analyses[22–26,35,43,46,55–58]. In this study, we focused on temporal changes in the efficacy of eight *Rps* genes (*Rps1a*, *Rps1b*, *Rps1c*, *Rps1d*, *Rps1k*, *Rps3a*, *Rps6*, *Rps7*) that have been used to describe isolate pathotypes in *P. sojae* pathotype surveys for decades[22]. In soybean production, *Rps1c* and *Rps1k* are currently the most commonly deployed genes available in cultivars for the management of PRR, while *Rps1a*, *Rps3a*, and *Rps6* are available to a lesser extent[6,13]. We did not include *Rps2*, *Rps3b*, *Rps3c*, *Rps4* and *Rps5* in this analysis, as not all studies used these resistance genes to characterize their *P. sojae* pathotype composition of field populations, and none of these *Rps* genes are used for

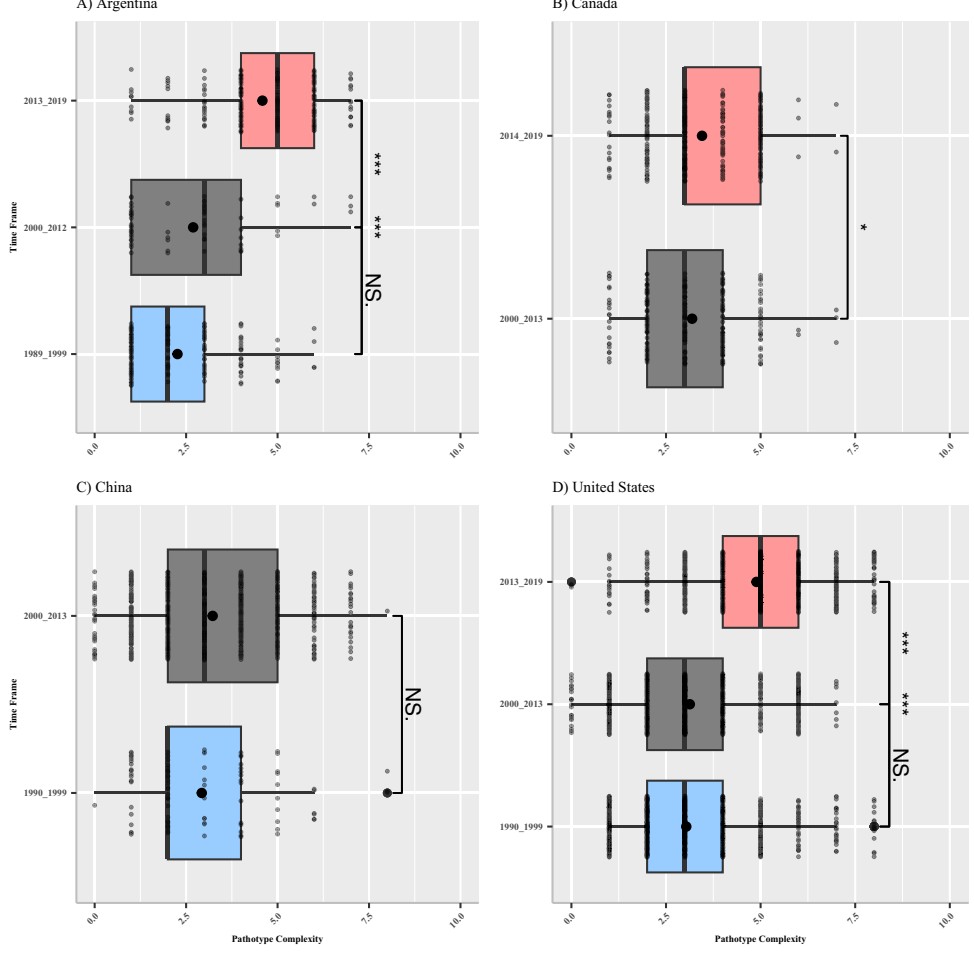

**Fig. 1 | Pathotype complexity of all isolates by timepoint for each country.** Panel **A** Argentina, 1989–1999 $n = 174$ isolates, 2000–2012 $n = 65$ isolates, 2013–2019 $n = 210$ isolates. 1989–1999 and 2000–2012 $t$-test ($t = -1.775$, df = 97.46, 95% CI = $-0.906$:0.051, $p = 0.079$), 2000–2013 and 2013–2019 $t$-test ($t = -7.9205$, df = 94.767, 95% CI = $-2.367$:$-1.418$, $p = 4.485 \times 10^{-12}$), 1989–1999 and 2013–2019 $t$-test ($t = -15.614$, df = 374.61, 95% CI = $-2.613$:$-2.029$, $p = <2.2 \times 10^{-16}$). Panel **B** Canada, 2000–2013 $n = 253$ isolates, 2014–2019 $n = 394$ isolates. 2000–2013 and 2014–2019 $t$-test ($t = -2.562$, df = 536.52, 95% CI = $-0.462$:$-0.061$, $p = 0.01067$). Panel **C** China, 1990–1999 $n = 97$ isolates, 2000–2013 $n = 790$ isolates. 1990–1999 and 2000–2013 $t$-test ($t = -1.545$, df = 119.9, 95% CI = $-0.678$:0.083, $p = 0.1249$). Panel **D** United States, 1990–1999 $n = 1115$ isolates, 2000–2013 $n = 956$ isolates, 2013–2019 $n = 1067$ isolates. 1990–1999 and 2000–2013 $t$-test ($t = -1.278$, df = 1998.1, 95% CI =

$-0.237$:0.05, $p = 0.2014$), 2000–2013 and 2013–2019 $t$-test ($t = -24.954$, df = 1894.5, 95% CI = $-1.89$:$-1.62$, $p < 2.2 \times 10^{-16}$), 1990–1999 and 2013–2019 $t$-test ($t = -28.021$, df = 2168.7, 95% CI = $-1.984$:$-1.725$, $p < 2.2 \times 10^{-16}$). Blue coloring denotes the studies performed in the 1990s (1989–1999, or 1990–1999), gray denotes the studies performed between 2000 and 2013, and red coloring denotes studies performed between 2013 and 2019 for each respective country. Dots indicate individual isolates pathotype complexity. Median pathotype complexity is depicted by the black bar within the box. Whiskers depict the first and third quartiles of data. Mean pathotype complexity is shown as a black circle within the boxplot. Outliers are defined as a black dot outside of the first and third quartiles. Asterisks indicate statistically significant differences between the means of groups at α = 0.05. Source data are provided as a Source Data file.

management of PRR in commercial soybean production in these nations.

## Pathotype complexity has significantly increased over time on a national scale

Increases in pathotype complexity within populations reflect the total number of *Rps* genes an individual *P. sojae* isolate within the population can cause disease against. Thus, these genes are ineffective for PRR management purposes. The published pathotype data from Argentina, Canada, China, and the United States were grouped into two or three distinct timeframes (generally: 1990–1999, 2000–2012, and 2013–2019) for all temporal analyses, depending on the availability of data from each country (see "Methods"; Table 1). The pathotype complexity of each tested *P. sojae* isolate was determined and summarized for each timeframe and country (Fig. 1). Differences in pathotype complexity between the 1990s and 2000–2013 timeframes within each country were not significant at $p = 0.05$, with observed increases in the mean complexity of less than 1 *Rps* gene (Fig. 1).

Significant increases in pathotype complexity over time were observed in Argentina ($t = -7.9205$, df = 94.767, 95% CI = $-2.367$:$-1.418$, $p = 4.485 \times 10^{-12}$), Canada ($t = -2.562$, df = 536.52, 95% CI = $-0.462$:$-0.061$, $p = 0.01067$), and the United States ($t = -24.954$, df = 1894.5, 95% CI = $-1.89$:$-1.62$, $p < 2.2 \times 10^{-16}$) between the 2000–2012/ 2013 and 2013–2019 timeframes (Fig. 1). The most notable shift was detected between the 2000–2012/2013 and 2013–2019 timeframes in Argentina and the United States, with a mean increase in population complexity of approximately 2 *Rps* genes (Fig. 1). In China, a small insignificant ($t = -1.545$, df = 119.9, 95% CI = $-0.678$:0.083, $p = 0.1249$) increase in pathotype complexity was observed between the two timeframes, with a mean increase of 0.317 *Rps* genes (Fig. 1).

## *Rps1c* and *Rps1k* are no longer effective for PRR management in Argentina, Canada, and the United States

Repeated use of an individual resistance gene to manage a phytopathogenic population over time can lead to a loss of management efficacy of that resistance gene[63,64]. In this study, an *Rps* gene is

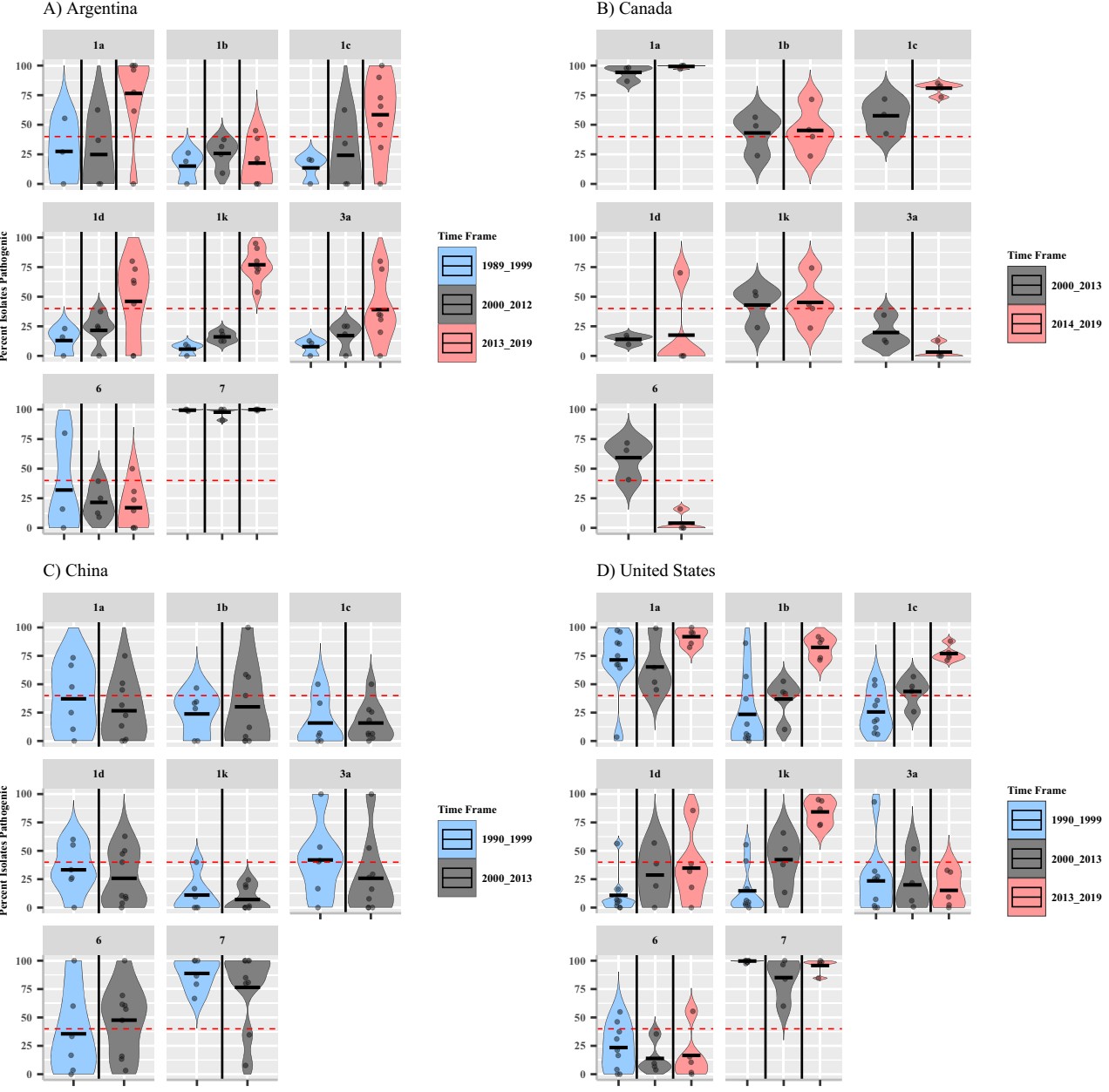

**Fig. 2 | Resistance gene efficacy for each *Rps* gene and timepoint interaction by country.** Facets denote the *Rps* genes tested; the *Y*-axis is the percent of isolates that are pathogenic on a given gene at a specific time frame from each study. Panel **A** Argentina, 1989–1999 *n* = 174 isolates, 2000–2012 *n* = 65 isolates, 2013–2019 *n* = 210 isolates. Panel **B** Canada, 2000–2013 *n* = 253 isolates, 2014–2019 *n* = 394 isolates. Panel **C** China, 1990–1999 *n* = 97 isolates, 2000–2013 *n* = 790 isolates. Panel **D** United States, 1990–1999 *n* = 1115 isolates, 2000–2013 *n* = 956 isolates, 2013–2019 *n* = 1067 isolates. Blue coloring denotes the studies performed in the 1990s (1989–1999, or 1990–1999), gray denotes the studies performed between 2000 and 2013, and red coloring denotes studies performed between 2013 and 2019 for each respective country. Dots within violin plots represent percent efficacy for each *Rps* gene by reported years isolates were recovered from included studies within each time frame. The black bar represents the mean percent pathogenic for each time frame and *Rps* gene within a county. The red dashed line is at 40%, indicating when the management efficacy of a gene to the population is reduced. Source data are provided as a Source Data file.

considered ineffective at managing a *P. sojae* population if ≥40% of the sampled population can evade detection and cause disease. This provides a conservative estimate of when new *Rps* genes should begin to be introduced to manage *P. sojae*. Here we determine if the efficacy of the primary *Rps* genes used for the management of PRR has been lost over time.

In Argentina, only *Rps7* was ineffective in managing the *P. sojae* population until the 2013–2019 time frame. Nearly 100% of isolates recovered in the first two timeframes were only pathogenic on *Rps7* (Fig. 2). However, isolates recovered from 2013–2019 were pathogenic on *Rps1a* (84.7%), *Rps1c* (66.6%), and *Rps1k* (78.1%). More than 75% of

the isolates tested were pathogenic on *Rps1a* and *Rps1k* (Fig. 2). Similarly, in the United States, *Rps1a* and *Rps7* were the only ineffective genes on a national scale until the 2013–2019 sampling timeframe. A greater percentage of isolates recovered during this third timeframe were pathogenic on *Rps1b* (83.5%), *Rps1c* (74.5%), and *Rps1k* (85.5%). Likewise, increases in pathogenicity between the 2000–2013 and 2013–2019 timeframes on soybean genotypes containing *Rps1a* (from 60.9 to 89.1%) and *Rps1b* (from 44.1 to 83.5%) were also observed. Interestingly, the efficacy of *Rps3a* and *Rps6* genes on the United States *P. sojae* national population appears to have remained consistent throughout the past thirty years (Fig. 2; Supplementary Fig. 1).

**Table 2 | Isolate number, unique pathotypes, mean pathotype complexity and Gleason diversity index for each population by timepoint**

| Country | Time frame | Isolate $n$ | Unique pathotypes | Mean pathotype complexity | Diversity index Gleason |
|---|---|---|---|---|---|
| United States | | | | | |
| | 1990–1999 | 1115 | 75 | 3.03 | 10.55 |
| | 2000–2013 | 933 | 77 | 3.12 | 11.11 |
| | 2013–2019 | 1064 | 80 | 4.89 | 11.33 |
| Argentina | | | | | |
| | 1989–1999 | 174 | 36 | 2.26 | 6.784 |
| | 2000–2012 | 65 | 18 | 2.69 | 4.072 |
| | 2013–2019 | 210 | 43 | 4.58 | 7.855 |
| Canada | | | | | |
| | 2000–2013 | 253 | 22 | 3.18 | 3.795 |
| | 2014–2019 | 394 | 36 | 3.45 | 5.856 |
| China | | | | | |
| | 1990–1999 | 96 | 30 | 2.92 | 6.354 |
| | 2000–2013 | 758 | 117 | 3.24 | 17.49 |

Source data are provided as a Source Data file.

Small differences in the *P. sojae* populations' pathogenicity on the *Rps1k* gene in Canada were observed between the 2000–2013 and 2014–2019 sampling timeframes, from 39.9 to 43.9% (Fig. 2; Supplementary Fig. 1). *Phytophthora sojae* population pathogenicity on the *Rps1c* gene in Canada increased substantially between sampling timeframes, with 54.1% of isolates being pathogenic on *Rps1c* in 2000–2013 compared to 83.7% of isolates recovered from 2014–2019 (Fig. 2; Supplementary Fig. 1). Similarly, there was an increase in the proportion of the population that was pathogenic on the *Rps1d* gene between 2000–2013 (13.4%) and 2014–2019 (52.5%), however, population pathogenicity on *Rps6* decreased from 55.7 to 11.9%. *Rps1d* is not widely deployed in Canadian soybean production[13]; it is curious why the sampled population is adapting to this resistance gene on a national scale. Of the *Rps* genes tested with the population from China, only the efficacy of *Rps6* changed between 1990–1999 (26.8%) and 2000–2013 (58.9%). The efficacy of the other tested *Rps* genes remained consistent throughout the timeframes used for analyses (Fig. 2; Supplementary Fig. 1). This could be because *P. sojae* is an invasive species to China and has only been present in that country since the early 1990s[47]. Thus, the *P. sojae* population in China may not be as genetically diverse or may have had less exposure to the tested resistance genes deployed compared with native North American populations.

### *Phytophthora sojae* pathotype diversity increases over time

As *P. sojae* populations adapt to resistance genes, differences in the pathogenic diversity among populations can be measured[65–67]. Here we investigate differences in pathotype diversity within and among temporally sampled international populations using the Gleason diversity index (see "Methods") and Principal Coordinates Analysis (PCoA), in tandem with testing beta-dispersion and Permutational Multivariate Analysis of Variance (PERMANOVA) to determine significant shifts in these populations over time. The Gleason diversity index exhibited an increase in pathotype diversity between the first and last timeframes for the *P. sojae* populations in the United States, Canada, and China (Table 2). In Argentina, diversity appeared to decrease between 1989–1999 and 2000–2012 before increasing again in 2013–2019, with an overall trend of increased pathotype diversity over time (Table 2). PCoA plots for each country revealed that the

national population at each sampling timeframe, while interspersed, had their own pathotype composition (Fig. 3). A test of beta-dispersion and PERMANOVA were performed on each country individually by sampling timeframe to determine if the differences observed in pathotype composition via the PCoA plots were significant. Significant changes in beta-dispersion were determined between all time frames within each country. In the United States, pathotypes became more dispersed between the 1990–1999 and 2000–2013 time frames (beta-dispersion mean difference = −0.145, $p$-value < 0.001; PERMANOVA $R^2 = 0.025$, $p$-value < 0.001), and began to coalesce during the 2013–2019 time frame as compared to the 2000–2013 time frame (beta-dispersion mean difference = 0.045, $p$-value < 0.001; PERMANOVA $R^2 = 0.119$, $p$-value < 0.001). Similar findings were obtained in Argentina between the 1989–1999 and 2000–2012 time frames (beta-dispersion mean difference = −0.144, $p$-values < 0.001; PERMANOVA $R^2 = 0.234$, $p$-value < 0.001), as well as the 2000–2012 and 2013–2019 time frames (beta-dispersion mean difference = 0.195, $p$-value < 0.001; PERMANOVA $R^2 = 0.059$, $p$-value < 0.001). Due to the availability of data, the change in pathotype beta-diversity could only be compared between 2000–2013 and 2014–2019 in Canada (beta-dispersion mean difference = 0.091, $p$-value < 0.001; PERMANOVA $R^2 = 0.086$, $p$-value < 0.001), and between the 1990–1999 and 2000–2013 time frame in China, of this study (beta-dispersion mean difference = −0.053, $p$-value < 0.001; PERMANOVA $R^2 = 0.018$, $p$-value < 0.001). However, the results from Canada and China corroborate what was found in the United States and Argentina during their respective time frame. Reductions in beta-dispersion over time indicate that more isolates are becoming similarly virulent on a number of the *Rps* genes tested. Beta-dispersion and PERMANOVA tests corroborated alpha-diversity test results; there was a significant difference in each country's pathotype composition between sampling timeframes, with pathotypes becoming less dispersed over time (Tables 3 and 4).

## Discussion

These analyses on a national scale support previous observations that pathotype compositions of *P. sojae* populations have changed over time, and pathotype complexity has increased[14,37–39]. This study demonstrates that *Rps* genes, such as *Rps1c* and *Rps1k*, have lost efficacy over time across major soybean-producing areas in the world, possibly because of the adaptive traits of *P. sojae* populations and the imposed selective pressure of frequently planting soybean varieties with the same *Rps* gene. Most importantly, this study reports that *Rps1c* and *Rps1k*, the most widely utilized *Rps* genes within the United States, Argentina, and Canada, are no longer effective for the management of PRR in these countries.

The data used in this systematic review was limited to the geographic areas sampled in *P. sojae* pathotype surveys conducted across the world over the past 30 years. As with any survey-based work, there are physical and financial limitations to the total area that could be sampled within each region and variations present within the methods used. Likewise, some countries did not perform these surveys in regular intervals, as was found with the absence of recent data from China, to be able to view the current efficacy of *Rps* genes. Similarly, we could not identify *P. sojae* pathotype studies conducted in Canada between 1990 and 1999.

The main potential biases in *P. sojae* pathotype surveys come from the method used to obtain isolates and inoculation conditions used for pathotype characterization. Isolations from field plants present an inherent bias on isolates recovered based on the plant genotype; only isolates that can cause disease on the present genotype are recovered. To account for this potential bias, field soil is collected, and a pathogen baiting technique is performed, which incites disease in a soybean genotype containing no known *Rps* genes. Subsequent isolation of *P. sojae* from those bait plants onto an oomycete selective media has become more regularly performed[68] (Table 1). Inoculation conditions,

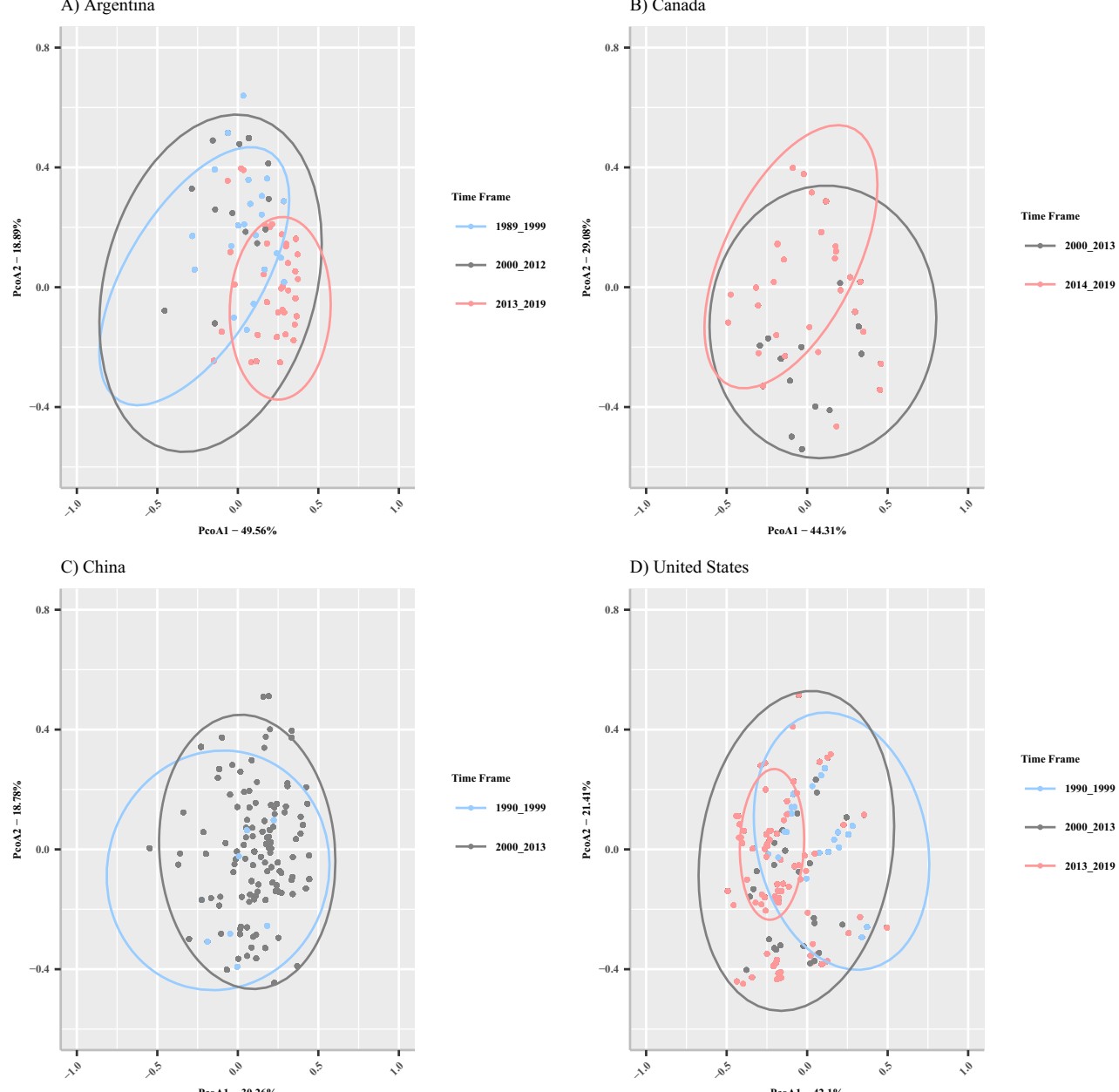

**Fig. 3 | Principal Coordinates Analysis (PCoA) of the sampled virulence phe-notype in *P. sojae* populations for each country colored by timepoint.** Panel **A** Argentina, 1989–1999 *n* = 174 isolates, 2000–2012 *n* = 65 isolates, 2013–2019 *n* = 210 isolates. Panel **B** Canada, 2000–2013 *n* = 253 isolates, 2014–2019 *n* = 394 isolates. Panel **C** China, 1990–1999 *n* = 97 isolates, 2000–2013 *n* = 790 isolates. Panel **D** United States, 1990–1999 *n* = 1115 isolates, 2000–2013 *n* = 956 isolates, 2013–2019 *n* = 1067 isolates Blue coloring denotes the studies performed in the 1990s (1989–1999 or 1990–1999), gray denotes the studies performed between 2000 and 2013, red coloring denotes studies performed between 2013 and 2019 for each respective country. Dots represent the Jaccard distance matrices values for each isolate within each country and time frame, respectively. 95% data ellipses are shown for each timepoint. Source data are provided as a Source Data file.

such as high temperature (>25 °C), can cause erroneous phenotypic results[68]. Of the 29 studies used in this systematic review, three used solely plant isolations, fifteen used solely soil baiting, and eleven used a combination of plant isolations and soil baiting to recover isolates of *P. sojae* (Table 1). This systematic review is also limited by the fact that plant isolation and soil baiting surveys were not separated for the analysis. All studies reported adequate inoculation conditions[11,13–16,27–34,36–42,44,45,48–54].

The uniform and significant increase in pathotype complexity and diversity over time is likely due to the selective pressure imposed by the frequent planting of soybean cultivars with single qualitative resistance genes over a large area. The 'Boom and Bust' cycle of resistance genes to phytopathogens is well documented in potatoes

and the pathogen *Phytophthora infestans*[64]. Resistance genes for management of late-blight of potato, caused by *P. infestans*, are typi-cally effective for only a few years before the pathogen population adapts, rendering them ineffective[64]. In our study, we identified that *Rps1c* and *Rps1k* were effective for PRR management for 20–30 years, which is longer than expected for R-gene-mediated management[11,69,70]. This may be due to biological characteristics specific to *P. sojae* as it,. S is homothallic, thus primarily self-fertilizing with rare occurrences of outcrossing with other genotypes; or also due to biases in the data used for the systematic review[59]. Additionally, *P. sojae* is a soilborne pathogen and thus, does not aerially disperse over long distances. These factors may contribute to the low genomic diversity observed within sampled *P. sojae* populations[38,71]. Even with limited genomic

## Table 3 | Tukey HSD results on Beta-dispersion comparisons

| Country | Timepoint | Mean difference | p-value |
|---|---|---|---|
| United States | | | |
| | 1990–1999/2000–2013 | −0.145 | <0.001 |
| | 1990–1999/2013–2019 | −0.101 | <0.001 |
| | 2000–2013/2013–2019 | 0.045 | <0.001 |
| Argentina | | | |
| | 1989–1999/2000–2012 | −0.144 | <0.001 |
| | 1989–1999/2013–2019 | 0.051 | <0.001 |
| | 2000–2012/2013–2019 | 0.195 | <0.001 |
| Canada | | | |
| | 2000–2013/2014–2019 | 0.091 | <0.001 |
| China | | | |
| | 1990–1999/2000–2013 | −0.053 | <0.001 |

Pairwise comparison of means using Tukey Highly Significant Differences multiple comparisons of means after significance identification with one-way ANOVA (United States: df = 2, sum sq = 11.589, mean sq = 5.7945, F value = 222.26, p = <2.2$^{-16}$; Argentina: df = 2, sum sq = 1.5516, mean sq = 0.77582, F value = 13.633, p = 1.791$^{-6}$; Canada: df = 1, sum sq = 2.0691, mean sq = 2.06909, F value = 196.59, p = <2.2$^{-16}$; China: df = 1, sum sq = 0.2382, mean sq = 0.238242, F value = 11.935, p = 0.0005779) on Beta-dispersion results. Source data are provided as a Source Data file.

## Table 4 | Pairwise PERMANOVA comparisons of country time frames

| Country | Time frame interaction | Significance | R² | Permutations |
|---|---|---|---|---|
| United States | | | | |
| | 1990–1999/2000–2013 | 0.001 | 0.025446 | 999 |
| | 1990–1999/2013–2019 | 0.001 | 0.023582 | 999 |
| | 2000–2013/2013–2019 | 0.001 | 0.118946 | 999 |
| Argentina | | | | |
| | 1989–1999/2000–2012 | 0.001 | 0.234361 | 999 |
| | 1989–1999/2013–2019 | 0.001 | 0.085718 | 999 |
| | 2000–2012/2013–2019 | 0.001 | 0.059392 | 999 |
| Canada | | | | |
| | 2000–2013/2014–2019 | 0.001 | 0.085729 | 999 |
| China | | | | |
| | 1990–1999/2000–2013 | 0.001 | 0.018057 | 999 |

Source data are provided as a Source Data file.

diversity, *P. sojae* pathotypes continue to evolve to be more diverse and pathogenic on an increasing number of soybean *Rps* genes. As the diversity and complexity of pathotypes within a *P. sojae* population increase, the likelihood that a single host resistance gene will be able to effectively manage that population decreases. Incorporating multiple qualitative resistance genes within soybean varieties will be necessary to broadly manage *P. sojae* populations via *Rps* gene-mediated resistance.

*P. sojae* is thought to be native to the United States as a pathogen of the indigenous legume genus *Lupinus*[59,71,72] and was first described shortly after soybean production started in the United States. *P. sojae* genotypes have been shown to be largely clonal, with most of the genetic variation within the species encompassed within four distinct genotypes[59,73]. Subsequent work in the United States and Argentina substantiated this work, identifying low to moderate genetic variation and evidence of potential sub-populations within the United States and therefore providing more evidence that the United States is the center of origin for *P. sojae*[38]. However, no correlation between genetic diversity and pathotype diversity within these populations has thus far been identified[45,59,74]. *P. sojae Avr* genes have been shown to be concentrated within transposon-rich regions of the genome, indicating

that the adaptive evolution of *Avr* gene sequences is more rapid than the transposon-sparse regions of the genome[75]. Due to this genomic arrangement, *Avr* gene sequences containing single nucleotide polymorphisms, insertions, and deletions have been identified within the coding region of the *P. sojae Avr*1a, *Avr*1b, *Avr*1c, *Avr*1d, *Avr*1k, *Avr*3a, and *Avr*6 genes subsequently conferring virulence on the corresponding soybean *Rps* gene[19]. Thirty-one typical isolates from the 2000–2013 survey in Canada were fully sequenced and were found to contain all haplotypes reported for seven *Avr* genes of P. sojae throughout the world[19]. In addition, the sequences of those genes were compared with ca. 300 isolates from the 2014–2019 survey and found a great increase in pathotype complexity from the earlier survey that could be linked to selection pressure from *Rps* usage in soybean fields over the years[13].

To our knowledge, *Rps1d* has never been deployed commercially anywhere in the world, and the gene has never been identified precisely or cloned. Surprisingly, we observed an increase in isolates pathogenic against the *Rps1d* gene. This may indicate that *Rps1d* has been deployed unwittingly in certain commercial soybean varieties, as has been previously reported for *Rps6*[76,77]. Additionally, previous studies have found a significant percentage of isolates with virulence on *Rps1d*[14,74], thus supporting the unknown presence of *Rps1d* in some soybean varieties. Other studies in Canada have found nearly no isolates virulent against *Rps1d*[41], suggesting that issues existed with differentials carrying *Rps1d*[13], a distinct possibility considering the elusive nature of the gene. The rise in virulence in the 2014–2019 survey would probably be explained by the fortuitous presence of *Rps1d* in many commercial lines, although this will only be answered by the eventual identification of *Rps1d*.

Adaptation of the *P. sojae* population to *Rps1c* and *Rps1k* took longer than anticipated, more than double the time originally predicted by Schmitthenner[11]. Research into the soybean–*P. sojae* molecular interactions has shown that *Rps1k* is able to detect and mount a defense response to both *Avr1k* and *Avr1b*[78]. A similar study showed *Rps1c* detected *Avr1c* and *Avr1a*[79]. This partially explains the longevity of *Rps1c* and *Rps1k* as effective PRR management tools, as both *Rps1c* and *Rps1k* would confer resistance to not only those *P. sojae* isolates expressing the *Avr1c* and *Avr1k* genes but also the *Avr1a* and *Avr1b* genes, respectively. In this study, the efficacy of *Rps1b* was shown to mirror that of *Rps1k* in Canada and the United States, even though *Rps1b* has never been knowingly deployed for PRR management in these countries (Fig. 2; Supplementary Fig. 1). This could be due to the *Rps1k* gene providing selection pressure against the *Avr1b* and *Avr1k* genes over time. *Rps1a* was already ineffective for management at the earliest time points used in this study for Canada and the United States. However, in Argentina and Canada, *Rps1a* and *Rps1c* had similar efficacy trajectories over time (Supplementary Fig. 1). *Rps6* was found to be ineffective in Canada during the 200–2013 sampling time frame and then effective during 2014–2019 samplings. This could be due to the increase in geographic sampling performed, and therefore an increase in the geographically dispersed *P. sojae* isolates during the 2014–2019 timepoint to include Manitoba and Quebec, and therefore more representative of the Canadian *P. sojae* population (Fig. 2).

While the loss of the *Rps1c* and *Rps1k* genes for management was gradual in the United States over the sampling timeframes investigated, the loss of efficacy of these genes occurred more quickly in Argentina between 2000–2013 and 2014–2019 (Fig. 2; Supplementary Fig. 1). Grijalba and Gally[15] hypothesized that the increase in pathotype complexity in Argentina between the 1990s and the early 2000s was due to a combination of *Rps* gene and conservation tillage adoption, along with the presence of the indigenous population of *P. sojae* present in soils which were never planted with soybean[80]. The first *Rps* genes introgressed into Argentina soybean varieties were *Rps1a* and *Rps1c* in 1984, followed by *Rps1k* in 1991[81]. By 1998 these *Rps* genes, along with conservation tillage practices, were widely adopted in

Argentina (P. Grijalba, *personal observation*). Following the hypothesis made by Grijalba and Gally[15], pathotype complexity and diversity continued to increase through the 2000s while effective resistance genes in the region decreased. Nevertheless, the loss of *Rps1c* and *Rps1k* for management of PRR in North and South American commercial soybean production presents an immediate concern for global food security.

Single-dominant qualitative resistance genes are the most economically viable option for disease management, as plants are completely resistant to the pathogen population. However, this imposes significant pressure on the phytopathogenic community, selecting for those individuals that can mitigate, manipulate, or evade the plant's ability to detect the pathogen. As we have shown, the pathotypes of *P. sojae* are diverse within populations, and single *Rps* gene-mediated management will not be an effective long-term management strategy going forward unless new and widely effective *Rps* genes are deployed into commercial soybean varieties on a regular basis. In Canada, it was found that 85% of soybean growers were using an ineffective *Rps* gene within their fields, showing the need for novel and effective *Rps* genes[13]. Reports from the United States estimate that effective *Rps* genes such as *Rps3a* and *Rps6* were present in less than 3% of soybean varieties from 2010 to 2020[37]. The efficacy of *Rps3a* and *Rps6* remaining relatively stable during the past thirty years in the United States is likely due in part to their very limited availability to producers and therefore minimal selective pressure on the *P. sojae* populations studied here (Fig. 2; Supplementary Fig. 1). There have been over 40 *Rps* genes discovered in soybean germplasm, however, novel *Rps* genes are rarely screened during traditional pathotype survey studies[12]. A promising new *Rps* gene, *Rps11*, was recently identified and conferred *Rps*-mediated resistance to 80% of *P. sojae* isolates tested within their study, including even highly diverse pathotypes[82]. The *Rps11* gene sequence and associated markers for introgression into commercial soybean varieties have been patented by Corteva Agrosciences™, which is a promising first step towards the *Rps11* gene being available in commercial varieties in the near future[82]. However, even with the added efficiency of marker-assisted breeding, it will likely take a few years before varieties with the *Rps11* gene are available. Increasing the availability of already positioned and effective *Rps* genes, such as *Rps3a* and *Rps6*, would benefit PRR management in the interim. A concerted, discipline-wide effort will be needed to amend the soybean *Rps* genes tested in pathotype surveys moving forward so that new effective *Rps* genes can be identified for introgression into elite soybean varieties for future deployment.

Quantitative resistance (also called "partial resistance", "horizontal resistance", or "field tolerance") to *P. sojae* is available in commercial soybean varieties[6,83–85]. Unlike qualitative resistance, quantitative resistance is governed by multiple genes acting in concert to provide partial resistance to the pathogen by allowing infection but limiting the growth and spread of the pathogen within the plant and limiting disease development. As there are many genes acting together, there is no single focal point of selective pressure for the *P. sojae* population to adapt and cause disease[86]. Currently, more than 130 quantitative trait loci (QTL) have been identified in soybean for resistance to *P. sojae*, along with their associated markers for marker-assisted breeding, which are available in Lin et al.[12].

A conclusion of this systematic review is the predominant *Rps* genes used for PRR management in major soybean-producing countries of North and South America, *Rps1c* and *Rps1k*, are no longer effective, and alternative forms of resistance will be needed to manage PRR going forward. Concentrating breeding efforts on introgressing quantitative resistance into commercial soybean varieties, in conjunction with stacking and deploying novel *Rps* genes, may reduce the impact of future epidemics of PRR on global food security in these countries.

## Methods

This review follows reporting criteria in accordance with the Preferred Reporting Items for Systematic Review and Meta-analysis[87]. This systematic review has not been registered in the International Prospective Register of Systemic Reviews (PROSPERO), and the protocol has not been implemented or published before. The PRISMA checklist for this manuscript can be found in the supplement (Supplementary Note 1).

### Pathotype survey data

*Phytophthora sojae* pathotype studies were identified using Google Scholar and Web of Science on the 21st of September 2021 and searched again for studies published after the original search date on the 6th of June 2022. Data from all identified *P. sojae* pathotype surveys were transcribed manually into Microsoft Excel® or supplied by authors from published manuscripts (Table 1). Each survey dataset was then validated using the 'hagis' R package for microbial phenotypic pathogenicity data to ensure the data collected accurately reflected what was reported in the published manuscripts[88]. Studies that were not regional survey-based *P. sojae* pathotype studies on *Rps* gene efficacy were not used within this temporal analysis. A curated global *P. sojae* virulence phenotype database was established from all identified studies as of June 6th, 2022, and published online[89].

### Data preparation and eligibility criteria

Using the global pathotype dataset available, Argentina, Canada, China, and the United States were identified to have pathotype survey studies conducted regularly or intermittently over time for this temporal analysis[11,13–16,27–34,36–42,44,45,48–54]. Studies from countries that performed minimal pathotype surveys over time, or were outside of the time frames studied, were not used in analysis[22–26,35,43,46,55–58]. Phenotypic interactions of each *P. sojae* isolate were first reduced to a subset of *Rps* genes used in this study (*rps* (susceptible control), *Rps1a*, *Rps1b*, *Rps1c*, *Rps1d*, *Rps1k*, *Rps3a*, *Rps6*, *Rps7*) to ensure homogeneity of phenotypic observations for each isolate. The *Rps7* data from Canada was excluded as the most recent survey used a molecular tool to detect *P. sojae* Avr genes, and *Rps7* testing was not performed[13]. All other pathotype surveys used the soybean hypocotyl inoculation technique for pathotype characterization, as described in Dorrance et al.[68]. *P. sojae* isolates data were then grouped by country and timeframes for analysis by the year each sample was reportedly isolated from soil or from symptomatic plants. Data from studies that were outside the discrete timeframes examined were not used in the analysis, and no isolate data was used in more than one timeframe (Table 1). Timeframes for each country and the number of isolates used in each time point for analyses were as follows: United States: 1990–1999 ($n = 1115$), 2000–2013 ($n = 956$), 2013–2019 ($n = 1067$) (Supplementary Fig. 2). Argentina: 1989–1999 ($n = 174$), 2000–2012 ($n = 65$), 2013–2019 ($n = 210$) (Supplementary Fig. 3). Canada: 2000–2013 ($n = 253$), 2014–2019 ($n = 394$) (Supplementary Fig. 4). China: 1990–1999 ($n = 97$), 2000–2013 ($n = 790$) (Supplementary Fig. 5).

### Temporal analysis of pathotype complexity

Using the 'hagis' R package, pathotype complexity (sum of tested *Rps* genes an isolate can cause disease against) was determined for each individual *P. sojae* isolate by country before calculating the mean pathotype complexity and standard deviation of a sampled *P. sojae* population at each timeframe and country. A Welch's two-sample, two-sided *t*-test was used to determine the significance between pathotype complexity at each timeframe by country.

### Temporal comparison of *Rps* gene efficacy

Pathotype data for each country was grouped by timeframe, and the number and percentage of isolates pathogenic on each *Rps* gene was calculated as follows: 'number of isolates pathogenic' = (sum of isolates pathogenic on each *Rps* gene); percent pathogenic =

(('number of isolates pathogenic')/(total isolate N))*100. Percent isolate pathogenic results were plotted using 'ggplot2'[90] (Version 3.3.5) with a dashed red bar at the 40% management efficacy threshold to show effective and ineffective *Rps* genes for each of the sampled population timeframes. Forty percent was used as a conservative estimate for management efficacy, as it is before the majority of the population would be able to cause disease against a given *Rps* gene.

### Temporal differences in pathotype diversity

**Alpha-diversity.** The Gleason diversity index, and number of unique pathotypes, were calculated for country timeframes using the 'hagis' R package[88]. The Gleason diversity index is calculated as: Gleason = ((N unique pathotypes) − 1)/log(number isolates)[91]. The 'hagis' R package automatically removes non-pathogenic phenotypic data from the dataset before calculating diversity indices so that only differences in pathogenicity are described.

**Beta-diversity.** To make statistical comparisons between pathotype diversity for each timeframe, the data from each country was further reduced to exclude *P. sojae* isolates which were non-pathogenic on all of the 8 *Rps* genes, and the susceptible (*rps*) control phenotype was removed. Jaccard distances were calculated for each country individually using the 'vegan' R package[92] (version 2.5–7). Principal Coordinates Analysis was performed using the Jaccard distance matrix for each country and visualized with 95% data ellipses from the 'ellipses' R package[93] (Version 0.4.2) using 'ggplot2'[90] (Version 3.3.5). Beta-dispersion tests were conducted between timeframes using the function "betadisper" from the 'vegan' package[92]. An Analysis of Variance (ANOVA) was used to identify significant differences in Beta-dispersion between timeframes, and the ad-hoc Tukey's HSD test was used to determine significant pairwise differences between groups. A Permutational Multivariate Analysis of Variance (PERMANOVA) was performed using "adonis", with 999 permutations, from the 'vegan' R package to test for significant differences in the pathotype composition of the sampled populations within each country between timeframes. PERMANOVA significance was further investigated using a pairwise PERMANOVA from the 'RVAideMemoire' R package[94] (Version 0.9-81-2) with the function "pairwise.perm.manova" with 999 permutations.

### Reporting summary

Further information on research design is available in the Nature Portfolio Reporting Summary linked to this article.

## Data availability

The pathotype data used in this study has been deposited into the Zenodo database under accession number 7850345[89] and can be found on GitHub (https://github.com/AGmccoy/Phytophthora-sojae-global-pathotype-meta-analysis). The data are available under the Creative Commons Zero v1.0 Universal License. Source data are provided with this paper.

## Code availability

The R code used in this study has been deposited into the Zenodo database under accession number 7850345[89] and can be found on GitHub (https://github.com/AGmccoy/Phytophthora-sojae-global-pathotype-meta-analysis). The R code are available under the Creative Commons Zero v1.0 Universal License.

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

## Acknowledgements

We thank the North Central Soybean Research Program, United Soybean Board, Michigan Soybean Committee, Project GREEEN—Michigan's plant agriculture initiative; USDA National Institute of Food and Agriculture, Hatch project 1025521 and Michigan AgBioResearch for partial support of this study.

## Author contributions

A.G.M. conceived the study idea, facilitated the collaboration, determined methods, performed data analysis, curated the *P. sojae* pathotype database, and wrote the manuscript. R.B. collected data from Ontario, Canada, and revised the manuscript with details on local disease management regimes. C.B. collected data from Kentucky used in this study. D.G.C.-G. collected data from Nebraska and revised the manuscript. V.C.G. collected data from Iowa and Illinois and revised the manuscript. L.J.G. collected data from Nebraska. P.E.G. collected data from Argentina and provided details on local disease management regimes. E.G. collected data from Argentina and provided details on local disease management regimes. M.A.H. collected data from Manitoba, Canada. Y.M.K. collected data from Manitoba, Canada, and provided details on local disease management regimes. D.K.M. collected data from Minnesota and revised the manuscript. R.L.M. collected data from Iowa. S.X.M. collected data from Illinois and revised the manuscript. Z.A.N. aided in the planning of methods and data analysis and revised the manuscript. A.E.R. collected data from Iowa, United States, and revised the manuscript. M.G.R. revised the manuscript and provided insight into plant-microbe interactions. C.L.S. collected data from Iowa, United States. D.L.S. provided insight into *P. sojae* in Wisconsin, United States, and revised the manuscript. A.H.S. reviewed the code used in the analysis, aided in producing the database and revised the manuscript. D.E.P.T. collected data from Indiana, United States, and revised the manuscript. V.T. collected data from Quebec, Ontario, and Manitoba, Canada. O.W. collected data from Manitoba, Canada, and revised the manuscript. M.I.C. helped plan the methods, provided feedback on the results and direction of the study, revised the manuscript and is the primary investigator.

## Competing interests

The authors declare no competing interests.
