## [Peer Review File · Nature Communications]

Reviewers' Comments:

Reviewer #1:

Remarks to the Author:

I think this is a novel and well-written manuscript with very important conclusions. In particular I thought the introduction introduced major gene resistance in plants to a wider scientific audience very well. My comments are brief, mostly regarding grammatical issues. With that said, I have one constructive comment: I strongly suggest you consult the PRISMA 2020 guidelines for systematic reviews and meta-analyses: <https://doi.org/10.1136/bmj.n71>

As far as I can see, your methods section are already in compliance with the guidelines, and it should be relatively easy to ensure you are in compliance and cite PRISMA. This ensures that articles referring to themselves as "meta-analyses" comply with basic guidelines.

41 recovered - consider an alternative word - conducted?

66 Billion -> billion

72 to -> for

73 conditions are conducive

96 I think for the Nature Communications audience, give "effectors" as a parenthetical synonym for Avr proteins. I see this is done later in the paragraph - I would either move it to the first mention of Avr

119-121 this sentence needs to be cleaned up, specifically "in which" and "disease on"

124 Also worth pointing out here that gene stacking also theoretically increases durability

128 international scale -> worldwide

130 across continents

140 how sampled *P. sojae* populations pathotype diversity -> how the pathotype diversity of sampled *P. sojae* populations

141 I am not used to seeing results included at the end of the introduction

162 8 -> eight

174 need to rephrase "cause disease on" - *P. sojae* is not causing disease on a gene, and "against" is probably better than "on"

201 same comment, and continued throughout

252 Meta-analyses results -> Meta-analysis

266 identified that

326-327 introgression or insertion - at least in the US, soy is often GMO

335 al.

336 I would refer to this manuscript as a single meta-analysis, but this is a editorial choice

358-359 did all studies use the same methods for path tests?

396 there is an extra) in your equation

397 non-pathogenic

398 only differences

It is not necessarily missing from the manuscript, but I think, considering the audience of the journal and novelty of the manuscript, that it would be worth include an additional brief discussion of what "increasing pathogen complexity" actually looks on the level of an individual pathogen genome, to the extent this is described in the literature. I don't know enough about the pathosystem, but if a single Avr gene isn't necessary for infection, then do the "most complex" isolates have many Avr genes missing from their genome or silenced? Or, do the "most complex" isolates have extra chromosomes or duplications to allow for multiple versions of a particular Avr?

Reviewer #2:

Remarks to the Author:

The manuscript entitled "A global-temporal meta-analysis perspective on *Phytophthora sojae* resistance-gene efficacy for disease management" described phenotypic pathogenicity data of 5121 *P. sojae* strain isolates from 29 surveys in United states, Argentina, Canada and China. All selected *P. sojae* strains were isolated in mentioned areas between 1990 and 2019. Based on this *P. sojae* population, authors examined temporal and national differences in pathotype diversity and identified effective RPS genes for further disease management. Authors indicated that Rps1a, Rps1c and Rps1k are no longer effective for disease control in USA, Argentina or Canada, and Rps3a, Rps6 or Rps11 would be useful in further commercial usages in such areas.

Whilst this work is important and novel, there are several points need to concerned. In particular, the observation and analysis as it currently stand does not provide a sufficient insight into global *P. sojae* disease management. Different regions may require different solution. Meanwhile, isolates from some countries are not updated to my knowledge so the conclusion may not be timely and sufficiently useful.

1. It would be better to have an overall geographical distribution of all tested *P. sojae* isolates, for example, like Figure 1 in '<https://doi.org/10.1094/PHYTO-12-20-0561-R>', which would greatly increase the readability in this work.

2. Authors have not presented any genotyping analysis in this work. I would expect to see some related data, such as genotyping by SSR markers in some typical isolates, which gives more details about phylogenetic relationships of *P. sojae* population in different areas.

3. In Line 218-220, authors indicated, "Rps1d is not widely deployed in Canadian soybean production, it is curious why the sampled population is adapting to this resistance gene on a national scale." Genotyping of *P. sojae* populations (2000-2013 and 2014-2019) from Canada might easily give some useful hints about the risen of virulences during these periods.

4. In Line 143-145, authors indicated, 'Rps1c and Rps1k are no longer effective in the USA, Argentina, and Canada; however there is little change in Rps1c and Rps1k efficacy observed in China.' Whilst the conclusions in USA, Argentina and Canada are convinced to me, it is hard to draw this conclusion with *P. sojae* strains in China which were isolated 10 years ago.

5. In figure 2, I am curious to the data of Rps6. In Argentina and USA, Rps6 looks effective through the past 30 years, while this resistance was not work in 2000-2013 but restored its effectiveness in 2014. It is important to discuss more about these data in this manuscript, as Rps6 was identified by Athow and Laviolette in 1982.

6. In figure 3, I would expect authors to display a PCA plot that contains pathotype data of strain isolates in all four countries, before they draw the conclusions in line 242-247.

Reviewer #3:

Remarks to the Author:

The paper describes a detailed survey of international literature on the effectiveness of soybean resistance genes against the pathogen, *P. sojae*. The paper is well written and an important contribution to an effort to raise awareness of the need to breed and disseminate new forms of resistance to this pathogen and is worthy of publication with relatively few suggested changes. I felt the script would benefit from some short considerations of the genetic diversity of *P. sojae* and how this relates to its virulence diversity. Could some of the differences between the findings in different countries relate to dissemination of the pathogen in trade of soybean and are there lessons that can be learned if that's the case? The origin of the pathogen is touched on in L296 where an indigenous population is mentioned in south America. Is the centre of origin known or relevant to this study as a risk of sources of resistance breaking strains?

Unsure about style of introducing the subject again in each of the results sections – it feels like one is going back to the introduction and that the text should be there and not in results. However, this is a stylistic comment and does not detract from the script.

Minor points

L7 typo in spelling of 'Vanessa'

L72 change to 'without the need of a second genotype'

L99 – 101 sentence is missing a verb. Perhaps 'are produced' after 'signals'?

L114 apostrophe incorrect in '1980's' since it is plural and not possessive or indicating missing letters, replace with '1980s'. Also, on L224, L294-5 and elsewhere if present.

L140 'sampled *P. sojae* populations pathotype diversity' possessive apostrophe needed in 'populations' pathotype diversity' but rewording to 'sampled pathotype diversity in *P. sojae* populations' would be better as less ambiguity. See also Line 693 and no doubt other places in the script.

L305 suggest 'economically viable option'

L315 change to singular 'germplasm'

L403 suggest 'non-pathogenic' rather than 'apathogenic'

L482 double .. in reference

L687 suggest elaboration to say 'show statistically significant differences between the means of groups at....'

REVIEWER COMMENTS

Reviewer #1 (Remarks to the Author):

I think this is a novel and well-written manuscript with very important conclusions. In particular I thought the introduction introduced major gene resistance in plants to a wider scientific audience very well. My comments are brief, mostly regarding grammatical issues. With that said, I have one constructive comment:

I strongly suggest you consult the PRISMA 2020 guidelines for systematic reviews and meta-analyses: <https://doi.org/10.1136/bmj.n71>

As far as I can see, your methods section are already in compliance with the guidelines, and it should be relatively easy to ensure you are in compliance and cite PRISMA. This ensures that articles referring to themselves as "meta-analyses" comply with basic guidelines.

Thank you for this suggestion. We have revised the manuscript (primarily lines 284-305, and lines 431-475), and Table 1, filled out the necessary PRISMA documentation (Manuscript and Abstract checklists, search strategies, risk of bias assessment for each study used), and have attached all files to conform to the PRISMA 2020 guidelines in the supplemental information.

Updated table 1 to include isolate recovery method (plant isolations or soil baiting) and pathotype evaluation method so the similarities and differences between studies is more clear. Columns "Isolate recovery method" and "Pathotype evaluation method" now include this information.

41 recovered - consider an alternative word - conducted?

Thank you, Conducted is a better fit for this sentence . This revision has been incorporated into the manuscript.

66 Billion -> billion

Thank you for this correction. The manuscript has been revised.

72 to -> for

Thank you for this correction, it has been implemented within the manuscript.

73 conditions are conducive

Thank you for catching this grammatical error, revisions have been implemented within the manuscript.

96 I think for the Nature Communications audience, give "effectors" as a parenthetical synonym for Avr proteins. I see this is done later in the paragraph - I would either move it to the first mention of Avr

The authors agree that "effector" should be introduced sooner in the manuscript. Therefore, the term is now introduced in the revised manuscripts line 104 as suggested.

119-121 this sentence needs to be cleaned up, specifically "in which" and "disease on"

The sentence (line 130) has been corrected for easier readability. Instances within the manuscript where isolates were described as "causing disease on" resistance genes have now been changed to "causing disease against". We believe this is more accurate for the interaction occurring. It now reads "...The number of *Rps* genes which isolates can cause disease against, as well as the diversity of pathotypes,..."

124 Also worth pointing out here that gene stacking also theoretically increases durability

Thank you for this comment. The authors agree that gene stacking should be introduced here rather than solely in the discussion. To that end, we have amended this sentence (lines 132-135 in revised manuscript) to introduce the idea before it is discussed:

As pathotype complexity of the *P. sojae* population increases, there will be fewer individual *Rps* genes that can effectively manage the known population and gene stacking, or having multiple *Rps* genes present within the genotype, will be needed for effective disease management.

128 international scale -> worldwide

The authors agree that that this is a more succinct description of the study's scope. This revision has been implemented within the revised manuscript line 142

130 across continents

Thank you for this grammatical correction. The revision has been implemented within the manuscript.

140 how sampled P. sojiae populations pathotype diversity -> how the pathotype diversity of sampled P. sojiae populations

Thank you for catching this grammatical error. We have revised this sentence, and others, where sentences needed to be rearranged to be grammatically correct.

141 I am not used to seeing results included at the end of the introduction

Thank you for this comment. We have read other Nature Communications manuscripts that follow this format and believe that it provides a quick overview for the readers on the most important results for this study. The authors would prefer to keep this formatting, so long as the Journal allows it.

162 8 -> eight

This correction has been implemented within the manuscript.

174 need to rephrase "cause disease on" - P. sojiae is not causing disease on a gene, and "against" is probably better than "on"

Thank you for pointing this out. The authors have fixed this phrasing within this sentence and others throughout the manuscript.

201 same comment, and continued throughout

Thank you, this has been implemented as per the line 174 revision.

252 Meta-analyses results -> Meta-analysis

Thank you for this edit. This was implemented in the revised manuscript.

266 identified that

Thank you for this grammatical edit, it has been implemented within the revised manuscript.

326-327 introgression or insertion - at least in the US, soy is often GMO

Thank you for this comment. While herbicide and insect resistance traits are inserted into the soybean genomes, *Rps*-genes are within the natural soybean genome and typically introgressed into new varieties through traditional breeding, and likely with Marker selected breeding if available to ensure the gene is present. Therefore, we have decided to leave the wording as it is.

335 al.

This oversight has been corrected within the manuscript.

336 I would refer to this manuscript as a single meta-analysis, but this is a editorial choice

Thank you for your comment. The authors agree that this is better stated as a single meta-analysis, rather than plural. This revision has been implemented within the manuscript.

358-359 did all studies use the same methods for path tests?

There was a single pathotype survey conducted in Canada "" (citation 13)that used a validated qPCR assay to determine the pathotype of isolates in that study. This was denoted in the Materials and Methods, under Data Preparation, lines 464-466: "The *Rps7* data from Canada was excluded as the most recent survey used a molecular tool to detect *P. sojae* Avr genes and *Rps7* testing was not performed (13)." We have added a sentence after this (line 466-467) which states: "All other pathotype surveys used the hypocotyl inoculation technique for pathotype characterization as described in Dorrance et al 2007 (68)"

396 there is an extra) in your equation

Thank you for seeing this. We have corrected the equation so that it is now correct within the revised manuscript.

397 non-pathogenic

Thank you for your input. "apathogenic" has now been changed to non-pathogenic throughout the revised manuscript.

398 only differences

Thank you for finding this grammatical error. It has been corrected within the revised manuscript.

It is not necessarily missing from the manuscript, but I think, considering the audience of the journal and novelty of the manuscript, that it would be worth include an additional brief discussion of what "increasing pathogen complexity" actually looks on the level of an individual pathogen genome, to the extent this is described in the literature. I don't know enough about the pathosystem, but if a single Avr gene isn't necessary for infection, then do the "most complex" isolates have many Avr genes missing from their genome or silenced? Or, do the "most complex" isolates have extra chromosomes or duplications to allow for multiple versions of a particular Avr?

Thank you for this suggestion. There are many different identified mechanisms of effector evasion within the *P. sojae* genome. To better describe the identified mechanisms of effector detection evasion, we have added the following text to the manuscript (lines 325-328).

P. sojae is thought to be native to the United States as a pathogen of the indigenous legume genus *Lupinus* (59, 71-72) and was first described shortly after soybean production started in the United States. *P. sojae* genotypes have been shown to be largely clonal, with most of the genetic variation within the species encompassed within four distinct genotypes (59, 73). Subsequent work in the United States and Argentina substantiated this work, identifying low to moderate genetic variation and evidence of potential sub-populations within the United States, and therefore providing more evidence that the United States is the center of origin for *P. sojae* (38). However, no correlation between genetic diversity and pathotype diversity within these populations has thus far been identified (59, 45, 74). *P. sojae* Avr genes have been shown to be concentrated within transposon rich regions of the genome, indicating that the adaptive evolution of Avr gene sequences are more rapid than the transposon sparse regions of the genome (75). Due to this genomic arrangement, Avr genes sequences containing single nucleotide polymorphisms, insertions, and deletions have been identified within the coding region of the *P. sojae* Avr1a, Avr1b, Avr1c, Avr1d, Avr1k, Avr3a, and Avr6 genes subsequently conferring virulence on the corresponding soybean *Rps* gene (19). Thirty-one typical isolates from the 2000-2013 survey in Canada were fully sequenced and were found to contain all haplotypes reported for seven Avr genes of *P. sojae*

throughout the world (19). In addition, the sequences of those genes were compared with ca. 300 isolates from the 2014-2019 survey and found a great increase in pathotype complexity from the earlier survey that could be linked to selection pressure from Rps usage in soybean fields over the years (13).

Reviewer #2 (Remarks to the Author):

The manuscript entitled "A global-temporal meta-analysis perspective on Phytophthora sojae resistance-gene efficacy for disease management" described phenotypic pathogenicity data of 5121 P. sojae strain isolates from 29 surveys in United states, Argentina, Canada and China. All selected P. sojae strains were isolated in mentioned areas between 1990 and 2019. Based on this P. sojae population, authors examined temporal and national differences in pathotype diversity and identified effective RPS genes for further disease management. Authors indicated that Rps1a, Rps1c and Rps1k are no longer effective for disease control in USA, Argentina or Canada, and Rps3a, Rps6 or Rps11 would be useful in further commercial usages in such areas.

Whilst this work is important and novel, there are several points need to concerned. In particular, the observation and analysis as it currently stand does not provide a sufficient insight into global P. sojae disease management. Different regions may require different solution. Meanwhile, isolates from some countries are not updated to my knowledge so the conclusion may not be timely and sufficiently useful.

1. It would be better to have an overall geographical distribution of all tested P. sojae isolates, for example, like Figure 1 in '<https://doi.org/10.1094/PHYTO-12-20-0561-R>2019;', which would greatly increase the readability in this work.

Thank you for this suggestion. The temporal sampling identified in the literature allows us to go down to a state or province sampling over time at best. We have made these figures, showing the spatial-temporal sampling of each country over time, however, the authors feel that they are cumbersome due to the spatial-temporal sampling interaction and should be added as supplemental files to accompany the manuscript. These figures have been labelled as Supplementary figures 2-5 and accompanying figure descriptions.

2. Authors have not presented any genotyping analysis in this work. I would expect to see some related data, such as genotyping by SSR markers in some typical isolates, which gives more details about phylogenetic relationships of *P. sojae* population in different areas.

Although the aim of this work focused on management efficacy, previous work from some co-authors have addressed the issue of genotyping. Thirty-one typical isolates from the 2000-2013 survey in Canada were fully sequenced and were found to contain all haplotypes reported for seven *Avr* genes of *P. sojae* throughout the world (Arsenault-Labrecque et al., 2018). In addition, the sequences of those genes were compared with ca. 300 isolates from the 2014-2019 survey and the authors found a great increase in pathotype complexity from the earlier survey that they could link to selection pressure from *Rps* usage in soybean fields over the years (Tremblay et al., 2021). The authors have included the broad findings and implications from the literature on this matter on lines 325-344:

P. sojae is thought to be native to the United States as a pathogen of the indigenous legume genus *Lupinus* (59, 71-72) and was first described shortly after soybean production started in the United States. *P. sojae* genotypes have been shown to be largely clonal, with most of the genetic variation within the species encompassed within four distinct genotypes (59, 73). Subsequent work in the United States and Argentina substantiated this work, identifying low to moderate genetic variation and evidence of potential sub-populations within the United States, and therefore providing more evidence that the United States is the center of origin for *P. sojae* (38). However, no correlation between genetic diversity and pathotype diversity within these populations has thus far been identified (59, 45, 74). *P. sojae Avr* genes have been shown to be concentrated within transposon rich regions of the genome, indicating that the adaptive evolution of *Avr* gene sequences are more rapid than the transposon sparse regions of the genome (75). Due to this genomic arrangement, *Avr* genes sequences containing single nucleotide polymorphisms, insertions, and deletions have been identified within the coding region of the *P. sojae Avr1a, Avr1b, Avr1c, Avr1d, Avr1k, Avr3a, and Avr6* genes subsequently conferring virulence on the corresponding soybean *Rps* gene (19). Thirty-one typical isolates from the 2000-2013 survey in Canada were fully sequenced and were found to contain all haplotypes reported for seven *Avr* genes of *P. sojae*

throughout the world (19). In addition, the sequences of those genes were compared with ca. 300 isolates from the 2014-2019 survey and found a great increase in pathotype complexity from the earlier survey that could be linked to selection pressure from Rps usage in soybean fields over the years (13).

3. In Line 218-220, authors indicated, "Rps1d is not widely deployed in Canadian soybean production, it is curious why the sampled population is adapting to this resistance gene on a national scale." Genotyping of P. sojae populations (2000-2013 and 2014-2019) from Canada might easily give some useful hints about the risen of virulences during these periods.

To our knowledge, *Rps1d* has never been deployed commercially anywhere in the world and the gene has never been identified precisely or cloned. Based on this premise, we agree that it was rather surprising to find many isolates virulent against this gene. This may indicate that *Rps1d* has been deployed unwittingly in certain commercial soybean varieties, as has been previously reported for *Rps6* (Athow and Laviolette 1982; Anderson et al., 2007). Additionally, previous studies have found a significant percentage of isolates with virulence on *Rps1d* (Stewart et al., 2015; Dorrance et al., 2016), thus supporting the unknown presence of *Rps1d* in some soybean varieties. Other studies in Canada have found nearly no isolates virulent against *Rps1d* (Henriquez et al. (2020)), suggesting that issues existed with differentials carrying *Rps1d* (Tremblay et al., 2021), a distinct possibility considering the elusive nature of the gene. In terms of genotyping, 31 isolates were fully genotyped from the 2000-2013 survey, with 30% being virulent on *Rps1d*. The rise in virulence in the 2014-2019 survey would probably be explained by the fortuitous presence of *Rps1d* in many commercial lines, although this will only be answered by the eventual identification of Rps1d.

Genotyping of isolates from these time periods may provide insight in to the presence or absence of these Avr genes within the genomes. However, the scope of this meta-analysis is focused on the efficacy of these *Rps* genes for disease management over time, and the phenotypic data showing these populations adapting over time. This information has been added to lines 345-355 in the manuscript :

To our knowledge, *Rps1d* has never been deployed commercially anywhere in the world and the gene has never been identified precisely or cloned. Surprisingly, we observed an increase in isolates pathogenic against the Rps1d gene. This may indicate that *Rps1d* has

been deployed unwittingly in certain commercial soybean varieties, as has been previously reported for *Rps6* (76; 77). Additionally, previous studies have found a significant percentage of isolates with virulence on *Rps1d* (74; 14), thus supporting the unknown presence of *Rps1d* in some soybean varieties. Other studies in Canada have found nearly no isolates virulent against *Rps1d* (41), suggesting that issues existed with differentials carrying *Rps1d* (13), a distinct possibility considering the elusive nature of the gene. The rise in virulence in the 2014-2019 survey would probably be explained by the fortuitous presence of *Rps1d* in many commercial lines, although this will only be answered by the eventual identification of *Rps1d*.

4. In Line 143-145, authors indicated, 'Rps1c and Rps1k are no longer effective in the USA, Argentina, and Canada; however there is little change in Rps1c and Rps1k efficacy observed in China.' Whilst the conclusions in USA, Argentina and Canada are convinced to me, it is hard to draw this conclusion with *P. sojae* strains in China which were isolated 10 years ago.

Thank you for this review. The authors agree that the phrasing used on these lines is ambiguous as to the time period in which there is no change. To correct this, we have amended the sentence to say “; however there is little change in Rps1c and Rps1k efficacy observed in China between the time frames studied...”. While the authors would like to include newer data from the soybean growing regions of China, there were no identified recent pathotype surveys in this region that we could use for this meta-analysis (line 287-291):

Likewise, some countries did not perform these surveys in regular intervals, as was found with the absence of recent data from China, to be able to view the current efficacy of *Rps* genes. Similarly, we could not identify *P. sojae* pathotype studies conducted in Canada between 1990 and 1999, though this does not detract from the current representation of studies in Canada.

5. In figure 2, I am curious to the data of Rps6. In Argentina and USA, Rps6 looks effective through the past 30 years, while this resistance was not work in 2000-2013 but restored its effectiveness in 2014. It is important to discuss more about these data in this manuscript, as Rps6 was identified by Athow and Laviolette in 1982.

Thank you for this comment. The authors have added to the discussion to recognize this odd interaction, which was not observed for other *Rps* genes in any of the countries used in this meta-analysis.

Lines 345-355:

To our knowledge, *Rps1d* has never been deployed commercially anywhere in the world and the gene has never been identified precisely or cloned. Surprisingly, we observed an increase in isolates pathogenic against the *Rps1d* gene. This may indicate that *Rps1d* has been deployed unwittingly in certain commercial soybean varieties, as has been previously reported for *Rps6* (76; 77). Additionally, previous studies have found a significant percentage of isolates with virulence on *Rps1d* (74; 14), thus supporting the unknown presence of *Rps1d* in some soybean varieties. Other studies in Canada have found nearly no isolates virulent against *Rps1d* (41), suggesting that issues existed with differentials carrying *Rps1d* (13), a distinct possibility considering the elusive nature of the gene. The rise in virulence in the 2014-2019 survey would probably be explained by the fortuitous presence of *Rps1d* in many commercial lines, although this will only be answered by the eventual identification of *Rps1d*.

Lines 370-374:

Rps6 was found to be ineffective in Canada during the 200-2013 sampling time frame and then effective during 2014-2019 samplings. This could be due to the increase in geographic sampling performed, and therefore an increase in the geographically dispersed *P. sojae* isolates, during the 2014-2019 timepoint to include Manitoba and Quebec, and therefore more representative of the Canadian *P. sojae* population (Figure 2).

6. In figure 3, I would expect authors to display a PCA plot that contains pathotype data of strain isolates in all four countries, before they draw the conclusions in line 242-247.

The authors are unsure of what the reviewer means in this case. Results of the

PCA (made of pathotype data of all strains used, from all countries, across time frames) were discussed in lines 256-271 and Figure 3 referenced for readers, and then displayed after the in text reference. The Authors would be happy to move the figure before this paragraph, but typically Figures are referenced in the text and then shown within manuscripts.

Reviewer #3 (Remarks to the Author):

The paper describes a detailed survey of international literature on the effectiveness of soybean resistance genes against the pathogen, *P. sojae*. The paper is well written and an important contribution to an effort to raise awareness of the need to breed and disseminate new forms of resistance to this pathogen and is worthy of publication with relatively few suggested changes.

I felt the script would benefit from some short considerations of the genetic diversity of *P. sojae* and how this relates to its virulence diversity. Could some of the differences between the findings in different countries relate to dissemination of the pathogen in trade of soybean and are there lessons that can be learned if that's the case? The origin of the pathogen is touched on in L296 where an indigenous population is mentioned in south America. Is the centre of origin known or relevant to this study as a risk of sources of resistance breaking strains?

Thank you for this suggestion. We have added to the manuscript to better describe the relation between genetic diversity and virulence diversity as is available within the literature. This was added to the manuscript, line 325-344

***P. sojae* is thought to be native to the United States as a pathogen of the indigenous legume genus *Lupinus* (59, 71-72) and was first described shortly after soybean production started in the United States. *P. sojae* genotypes have been shown to be largely clonal, with most of the genetic variation within the species encompassed within four distinct genotypes (59, 73). Subsequent work in the United States and Argentina substantiated this work, identifying low to moderate genetic variation and evidence of potential sub-populations within the United States, and therefore providing more evidence that the United States is the center of origin for *P. sojae* (38). However, no correlation between genetic diversity and pathotype diversity within these populations has thus far been identified (59, 45, 74). *P. sojae* *Avr* genes have been shown to be concentrated within transposon rich regions of the genome, indicating that the adaptive evolution of *Avr* gene sequences are more rapid than**

the transposon sparse regions of the genome (75). Due to this genomic arrangement, *Avr* genes sequences containing single nucleotide polymorphisms, insertions, and deletions have been identified within the coding region of the *P. sojae* *Avr1a*, *Avr1b*, *Avr1c*, *Avr1d*, *Avr1k*, *Avr3a*, and *Avr6* genes subsequently conferring virulence on the corresponding soybean *Rps* gene (19). Thirty-one typical isolates from the 2000-2013 survey in Canada were fully sequenced and were found to contain all haplotypes reported for seven *Avr* genes of *P. sojae* throughout the world (Arsenault-Labrecque et al., 2018). In addition, the sequences of those genes were compared with ca. 300 isolates from the 2014-2019 survey and found a great increase in pathotype complexity from the earlier survey that could be linked to selection pressure from *Rps* usage in soybean fields over the years (Tremblay et al., 2021).

Unsure about style of introducing the subject again in each of the results sections – it feels like one is going back to the introduction and that the text should be there and not in results. However, this is a stylistic comment and does not detract from the script.

Due to the length and multiple analyses used in this meta-analysis the authors believe that briefly reintroducing the pertinent subject of each main results section is beneficial to readers. The brief text used at the beginning of each results sub-section is to provide readers with contextual background on the results sub-section they are in.

Minor points

L7 typo in spelling of 'Vanessa'

Thank you for finding this error. I have double checked the spelling of this authors name and corrected it within the revised manuscript.

L72 change to 'without the need of a second genotype'

This error has been corrected within the revised manuscript. (changing "to" to "of")

L99 – 101 sentence is missing a verb. Perhaps 'are produced' after 'signals'?

Thank you for this grammatical revision. We have added "are produced" after signals so the sentence is complete.

L114 apostrophe incorrect in '1980's' since it is plural and not possessive or indicating missing letters, replace with '1980s'. Also, on L224, L294-5 and elsewhere if present.

Thank you for finding this error. The authors have reworded sentences which were using the possessive tense so that they are not using them anymore. Replaced "19X0's" with "19X0s" throughout the manuscript where appropriate.

L140 'sampled P. sojae populations pathotype diversity' possessive apostrophe needed in 'populations' pathotype diversity' but rewording to 'sampled pathotype diversity in P. sojae populations' would be better as less ambiguity. See also Line 693 and no doubt other places in the script. (done, but a bit confused)

Thank you for this revision. We did find instances of this error in other places in the manuscript and those sentences have been revised to correct this mistake.

L305 suggest 'economically viable option'

The authors agree. "...economic option..." has been changed to "...economically viable option..." in the revised manuscript.

L315 change to singular 'germplasm'

Thank you for this revision. "germplasms" has been changed to "germplasm" within the revised manuscript.

L403 suggest 'non-pathogenic' rather than 'apathogenic'

Thank you for your input. "apathogenic" has now been changed to non-pathogenic throughout the revised manuscript.

L482 double .. in reference

Thank you for identifying this error. The extra “.” Has been removed from this reference in the revised manuscript, and all other citations checked for occurrences of this error.

L687 suggest elaboration to say ‘show statistically significant differences between the means of groups at...’

Thank you for this revision. The sentence has been revised from “ Asterisks show significance between groups mean at...” to “Asterisks show statistically significant differences between the means of groups at...”. We believe this will provide clarity to readers on the significance testing.

** See Nature Portfolio’s author and referees' website at www.nature.com/authors for information about policies, services and author benefits.

Reviewers' Comments:

Reviewer #2:

Remarks to the Author:

I read through the manuscript and the response letter. My major concerns had been solved. It is indeed very hard to trace the very details and find the solid reasons. This meta analysis will be useful for global soybean breeding and disease management.

Reviewer #3:

Remarks to the Author:

I thank the authors for their positive and detailed responses to the suggestions of reviewers and recommend the revised script is published with no further changes required.

Reviewer #4:

Remarks to the Author:

I was asked to review this manuscript specifically with regard to the quantitative methods and meta-analysis, so I will restrict my comments only to those aspects of the manuscript.

First and foremost, I would not call this study a meta-analysis. There is a tendency by some to call any study which uses published--rather than primary--data a "meta-analysis", but that is an overly broad definition that is not commonly accepted within the meta-analysis community. This study absolutely has the elements of a systematic review, but does not include what I would consider the fundamental aspects of a meta-analysis (one example would be that meta-analysis almost always includes the degree of uncertainty across different studies in the analysis through some form of weighting). To be clear this is a nitpickery over the term as used in the manuscript and not a reflection of the quality of work. However, improper designation of the methodological approach does not help for clarity within the literature.

Another comment has to do with the search terms used to find studies to include in the analysis (Supplementary Note 1). According to that note, searches were done in both Web of Science and Google Scholar, with the following terms:

"Phytophthora sojae pathotype"

"Phytophthora sojae pathotype survey"

"Phytophthora sojae virulence survey"

"Phytophthora sojae Race"

Given the way it is written I am assuming that the quotations were included, thus restricting the search to those exact four phrases (I should note that if this is the case, the second search can only produce results that are a subset of the first).

As it seemed potentially restrictive to search for those exactly as stated, I quickly tried a number of variants in both Web of Science and Google Scholar, focusing on how words were combined in phrases, as well as abbreviating the genus. The following show the counts of results of each, respectively, clustered by the original search term:

"Phytophthora sojae pathotype" (2 WOS / 27 GS)

"P. sojae pathotype" (0 / 27)

"Phytophthora sojae" pathotype (33 / 1200)

"P. sojae" pathotype (26 / 662)

"Phytophthora sojae pathotype survey" (0 / 0)

"P. sojae pathotype survey" (0 / 1)

"Phytophthora sojae" "pathotype survey" (1 / 5)

"Phytophthora sojae" pathotype survey (7 / 1140)

"P. sojae" "pathotype survey" (1 / 4)
"P. sojae" pathotype survey (7 / 638)

"Phytophthora sojae virulence survey" (0 / 0)
"P. sojae virulence survey" (0 / 0)
"Phytophthora sojae" "virulence survey" (0 / 6)
"Phytophthora sojae" virulence survey (13 / 6700)
"P. sojae" "virulence survey" (0 / 1)
"P. sojae" virulence survey (11 / 3430)

"Phytophthora sojae Race" (6 / 149)
"P. sojae Race" (17 / 265)
"Phytophthora sojae" Race (212 / 4980)
"P. sojae" Race (160 / 2520)

Personally I find Google Scholar data to be difficult-to-almost-useless for this type of study (the lack of an API for convenient downloading of information, the redundancy due to different versions of the same article coming up as separate hits, and the overly broad definition of what makes up the "scholarly" part of the web), although one cannot deny that it is often able to identify more obscure literature than WOS. More power to the authors for attempting to use it. But even just focusing on the Web of Science output, the restriction to complete phrases as well as not searching for abbreviated genus/species names (particularly in combination with the rest of the phrase) may have restricted their search for input studies more than they anticipated.

I do not know that they missed any key literature, I am simply pointing out a possible limitation of the search method.

For what it's worth, I also ran these searches without any quotation marks at all, allowing the words to be treated as independent order rather than order-dependent queries. The results were:

Phytophthora sojae pathotype (33 / 1390)
P. sojae pathotype (27 / 1510)

Phytophthora sojae pathotype survey (7 / 1320)
P. sojae pathotype survey (7 / 1430)

Phytophthora sojae virulence survey (13 / 8890)
P. sojae virulence survey (13 / 2950)

Phytophthora sojae Race (251 / 6960)
P. sojae Race (185 / 7730)

For Google Scholar these likely lack enough specificity to be useful, while I suspect most of the WOS results are identical to those in the partially restricted searches I did above where I kept the genus and species linked (either as *Phytophthora sojae* or *P. sojae*).

Beyond these specific comments, the rest of the methods seem fine; I just would not call it a meta-analysis. It is a systemically driven study of already published data, and is certainly quantitative, but is only arguably a synthesis, and does not include the general methods or approaches common to meta-analysis.

REVIEWER COMMENTS

Reviewer #2 (Remarks to the Author):

I read through the manuscript and the response letter. My major concerns had been solved. It is indeed very hard to trace the very details and find the solid reasons. This meta analysis will be useful for global soybean breeding and disease management.

Thank you for this review, and your time for providing reviews on previous versions of this manuscript.

Reviewer #3 (Remarks to the Author):

I thank the authors for their positive and detailed responses to the suggestions of reviewers and recommend the revised script is published with no further changes required.

Thank you for this review, and your time for providing reviews on previous versions of this manuscript.

Reviewer #4 (Remarks to the Author):

I was asked to review this manuscript specifically with regard to the quantitative methods and meta-analysis, so I will restrict my comments only to those aspects of the manuscript.

First and foremost, I would not call this study a meta-analysis. There is a tendency by some to call any study which uses published--rather than primary--data a "meta-analysis", but that is an overly broad definition that is not commonly accepted within the meta-analysis community. This study absolutely has the elements of a systematic review, but does not include what I would consider the fundamental aspects of a meta-analysis (one example would be that meta-analysis almost always includes the degree of uncertainty across different studies in the analysis through some form of weighting). To be clear this is a nitpickery over the term as used in the manuscript and not a reflection of the quality of work. However, improper designation of the methodological approach does not help for clarity within the literature.

Thank you for making this distinction. Upon reviewing the criteria of meta-analyses and systematic reviews the authors agree. Specifically, systematic reviews provide evidence for defined research questions using published data that meet the eligibility criteria specified. This is what was performed in this study and all references to "meta-analysis" have been changed to either "systematic review" or simply "analyses" when

appropriate. This change is reflected in the Title, the manuscript itself, and associated tables, figures and supplementary information.

Another comment has to do with the search terms used to find studies to include in the analysis (Supplementary Note 1). According to that note, searches were done in both Web of Science and Google Scholar, with the following terms:

"Phytophthora sojae pathotype"
"Phytophthora sojae pathotype survey"
"Phytophthora sojae virulence survey"
"Phytophthora sojae Race"

Given the way it is written I am assuming that the quotations were included, thus restricting the search to those exact four phrases (I should note that if this is the case, the second search can only produce results that are a subset of the first).

As it seemed potentially restrictive to search for those exactly as stated, I quickly tried a number of variants in both Web of Science and Google Scholar, focusing on how words were combined in phrases, as well as abbreviating the genus. The following show the counts of results of each, respectively, clustered by the original search term:

"Phytophthora sojae pathotype" (2 WOS / 27 GS)
"P. sojae pathotype" (0 / 27)
"Phytophthora sojae" pathotype (33 / 1200)
"P. sojae" pathotype (26 / 662)

"Phytophthora sojae pathotype survey" (0 / 0)
"P. sojae pathotype survey" (0 / 1)
"Phytophthora sojae" "pathotype survey" (1 / 5)
"Phytophthora sojae" pathotype survey (7 / 1140)
"P. sojae" "pathotype survey" (1 / 4)
"P. sojae" pathotype survey (7 / 638)

"Phytophthora sojae virulence survey" (0 / 0)
"P. sojae virulence survey" (0 / 0)
"Phytophthora sojae" "virulence survey" (0 / 6)
"Phytophthora sojae" virulence survey (13 / 6700)
"P. sojae" "virulence survey" (0 / 1)

"P. sojae" virulence survey (11 / 3430)

"Phytophthora sojae Race" (6 / 149)

"P. sojae Race" (17 / 265)

"Phytophthora sojae" Race (212 / 4980)

"P. sojae" Race (160 / 2520)

Personally I find Google Scholar data to be difficult-to-almost-useless for this type of study (the lack of an API for convenient downloading of information, the redundancy due to different versions of the same article coming up as separate hits, and the overly broad definition of what makes up the "scholarly" part of the web), although one cannot deny that it is often able to identify more obscure literature than WOS. More power to the authors for attempting to use it. But even just focusing on the Web of Science output, the restriction to complete phrases as well as not searching for abbreviated genus/species names (particularly in combination with the rest of the phrase) may have restricted their search for input studies more than they anticipated.

Thank you for catching this. The quotations used in Supplementary Note 1 were there purely to distinguish the term strings used in the search within the documentation. The quotations were not used in Google Scholar or Web of Science when identifying literature and data for use in the systematic review. We now realize that this could be taken as the quotations being used for searching in Web of Science and Google Scholar. This error has been corrected in Supplementary Note 1 via the removal of the quotations.

The authors also agree with reviewer 4 that Google Scholar produces a plethora of literature to go through and was not the most efficient method to identify literature. However, the first search for literature was conducted during COVID lockdown measures, and thus Dr. McCoy had sufficient time to go through Google results and identify obscure studies which met our defined literature criteria. Citations within the identified literature were used to corroborate findings in Google Scholar and Web of Science to provide confidence that all the previously published data was identified.

I do not know that they missed any key literature, I am simply pointing out a possible limitation of the search method.

For what it's worth, I also ran these searches without any quotation marks at all, allowing the words to be treated as independent order rather than order-dependent queries. The results were:

Phytophthora sojae pathotype (33 / 1390)
P. sojae pathotype (27 / 1510)

Phytophthora sojae pathotype survey (7 / 1320)
P. sojae pathotype survey (7 / 1430)

Phytophthora sojae virulence survey (13 / 8890)
P. sojae virulence survey (13 / 2950)

Phytophthora sojae Race (251 / 6960)
P. sojae Race (185 / 7730)

For Google Scholar these likely lack enough specificity to be useful, while I suspect most of the WOS results are identical to those in the partially restricted searches I did above where I kept the genus and species linked (either as *Phytophthora sojae* or *P. sojae*).

Beyond these specific comments, the rest of the methods seem fine; I just would not call it a meta-analysis. It is a systemically driven study of already published data, and is certainly quantitative, but is only arguably a synthesis, and does not include the general methods or approaches common to meta-analysis.

Thank you for this review, as well as your time on reviewing this manuscript.